



# Evaluation of Historic and Operational Satellite Radar Altimetry Missions for Constructing Consistent Long-term Lake Water Level Records

Song Shu[1*], Hongxing Liu[2*], Richard A. Beck[3], Frédéric Frappart[4], Johanna Korhonen[5],Minxuan
Lan[3], Min Xu[6], Bo Yang[7], Yan Huang[8]

[1]Department of Geography and Planning, Appalachian State University, Boone, NC 28608, USA
[2]Department of Geography, the University of Alabama, Tuscaloosa, AL 35487, USA
[3]Department of Geography and Geographic Information Science, University of Cincinnati, Cincinnati, OH 45221, USA
[4]Laboratoire d'Etudes en Géophysique et Océanographie Spatiales (LEGOS), UMR 5566, CNRS/IRD/UPS, OMP, 14 Avenue Édouard Belin, 31400 Toulouse, France
[5]Finnish Environment Institute, SYKE, Freshwater Center, Latokartanonkaari 11, 00790 Helsinki, Finland
[6]College of Marine Science, University of South Florida, St. Petersburg, FL 33701, USA
[7]Department of Sociology, University of Central Florida, Orlando, FL 32816, USA
[8]Key Lab. of Geographical Information Science, Ministry of Education, School of Geographical Science, East China Normal University, Shanghai 200241, China

*Correspondence to*: Song Shu (shus@appstate.edu), Hongxing Liu (Hongxing.Liu@ua.edu)

**Abstract.** Thirteen satellite missions have been launched since 1985, with different types of radar altimeters onboard. This study intends to make a comprehensive evaluation of historic and currently operational satellite radar altimetry missions for lake water level retrieval over the same set of lakes and to develop a strategy for constructing consistent long-term water level records for inland lakes at global scale. The lake water level estimates produced by different retracking algorithms (retrackers) of the satellite missions were compared with the gauge measurements over twelve lakes in four countries. The performance of each retracker was assessed in terms of the data missing rate, the correlation coefficient r, the bias, and the Root Mean Square Error (RMSE) between the altimetry-derived lake water level estimates and the concurrent gauge measurements. The results show that the model-free retrackers (e.g. OCOG/Ice-1/Ice) outperform the model-based retrackers for all missions, particularly over small lakes. Among the satellite altimetry missions, Sentinel-3 gave the best results, followed by SARAL. ENVISat has slightly better lake water level estimates than Jason-1 and -2, but its data missing rate is higher. For small lakes, ERS-1 and ERS-2 missions provided more accurate lake water level estimates than Topex/Poseidon mission. In contrast, for large lakes Topex/Poseidon is a better option due to its lower data missing rate and shorter repeat cycle. GeoSat and GeoSat Follow-On (GFO) both have extremely high data missing rate. Although several contemporary radar altimetry missions provide more accurate lake level estimates than GFO, GeoSat was the sole radar altimetry mission between 1985 and 1990 that provided the lake water level estimates. With a full consideration of the performance and the operational duration, the best strategy for constructing long-term lake water level records should be a two-step bias





correction and normalization procedure. In the first step, use Jason-2 as the initial reference to estimate the systematic biases
with Topex/Poseidon, Jason-1 and Jason-3 and then normalize them to form a consistent Topex/Poseidon-Jason series. Then,
use Topex/Poseidon-Jason series as the reference to estimate and remove systematic biases with other radar altimetry
missions to construct consistent long-term lake water level series for ungauged lakes.

## 1 Introduction

About three percent of Earth's land surface is covered by lakes (Pekel et al., 2016). These lakes are the habitats for a great
number of aquatic and terrestrial species (Schindler and Scheuerell, 2002). They are also the major freshwater sources for
various human activities (Postel et al., 1996). The long-term variations of lake water levels were identified as sentinel for
climate change (Adrian et al., 2009;Williamson et al., 2009). The lake water level change can also have significant
influences on the local ecosystem and environment, e.g. the breeding success of fishes (Probst et al., 2009), the drainage of
thaw lakes (Pohl et al., 2009;Marsh et al., 2009;Jones and Arp, 2015) and landslide at lake coastal areas (Tyszkowski et al.,
2015). Monitoring lake water levels is important for a better understanding of their impact on the environment and for the
wise management of freshwater resources.

    At present, only a very small portion of lakes are monitored by gauge stations. The number of gauged lakes has decreased in
recent years owing to the high cost of installation and maintenance of gauge stations (Hannah et al., 2011). The
overwhelming majority of the lakes on Earth remain ungauged, particularly those located in remote areas with harsh
environments, e.g. the Arctic and the sub-Arctic regions. Many previous studies show that the lakes in these remote areas
have been experiencing dramatic changes with regard to the lake water balance (Turner et al., 2014), the timing and
magnitude of spring/early-summer flooding (Rokaya et al., 2018), and the lake ice cover phenology (Surdu et al., 2014), due
to the rapid climate warming (Karl et al., 2015). There is an urgent need to develop an alternative approach for effective
monitoring of lake water levels at the global scale.

    Satellite radar altimeters make surface elevation measurements by tracking the satellite orbit position and the range between
the satellite and the Earth's surface at nadir direction. They have been used widely to monitor lake water levels since the
1980s. Thirteen satellite missions have been launched with different radar altimeters onboard in the past three decades.
Those include the Geodetic/Geophysical Satellite (GeoSat, 1985 – 1989) and GeoSat Follow-on (GFO, 1998 – 2008)
developed by U.S. Navy, ERS-1 (1991 – 2000), ERS-2 (1995 – 2011), ENVISat (2002 – 2012), Cryosat-2 (2010 – present)
and Sentinel-3 (2016 – present) developed by ESA (European Space Agency), the Satellite with ARgos and ALtika
(SARAL, 2013 – present) developed jointly by CNES (Centre National d'Etudes Spatiales, the French Space Agency) and
ISRO (Indian Space Research Organization), the Topex/Poseidon (T/P, 1992 – 2005), Jason-1 (2002 – 2013), Jason-2 (2008
– present) and Jason-3 (2016 – present) developed jointly by NOAA (National Oceanic and Atmospheric Administration),
NASA (National Aeronautics and Space Administration), CNES and EUMETSAT (European Organization for the
Exploitation of Meteorological Satellites), and the HY-2A (2011 – present) developed by CNSA (China National Space



Administration). Most of the radar altimeters operate in a conventional low-resolution mode (LRM), whereas Sentinel-3 and Cryosat-2 operate in Synthetic Aperture Radar (SAR) mode.

Based on the elevation measurements collected by different satellite radar altimeters, five online databases have also been
developed to offer the time series of altimetry-derived water level estimates for major inland lakes around the world. These include the Hydroweb database (http://www.legos.obs-mip.fr/soa/hydrologie/hydroweb/) developed by the Laboratoire d'Etudes en Géophysique et Océanographie Spatiales (LEGOS) (Crétaux et al., 2011), the River and Lake database (http://www.cse.dmu.ac.uk/EAPRS/products_riverlake.html) built by the ESA and De Montfort University (ESA-DMU), the Global Reservoir and Lake Monitor (GRLM, https://ipad.fas.usda.gov/cropexplorer/global_reservoir/) developed by the
Foreign Agricultural Service of the United States Department of Agriculture (USDA) (Birkett et al., 2011), the Hydrosat developed by the Institute of Geodesy from the University of Stuttgart (http://hydrosat.gis.uni-stuttgart.de), and the Database for Hydrological Time Series over Inland Waters (DAHITI, https://dahiti.dgfi.tum.de/en/) launched by the Deutsches Geodätisches Forschungsinstitut der Technischen Universität München (DGFI-TUM) in 2013 (Schwatke et al., 2015b). The time series of water level estimates in these databases are produced by merging the elevation measurements from multiple
satellite radar altimeters with different processing strategies (Birkett and Beckley, 2010;Ričko et al., 2012;Schwatke et al., 2015b).

For each satellite radar altimeter, one or more dedicated algorithms have been designed to retrieve the surface elevations. Each algorithm is often designed to handle one type of Earth's surface. These radar altimetry algorithms are also known as retracking algorithms or simply *retrackers*. For example, there are four different retrackers designed for ENVISat altimeter,
including the *Ocean retracker* for ocean open water surface, the *Ice1 retracker* for general continental ice sheet surface, the *Ice2 retracker* for continental internal flat ice surface, and the *Sea-Ice retracker* for ocean ice surface (Frappart et al., 2006).

Many previous studies have evaluated different satellite radar altimeters in the retrieval of water levels over inland lakes with different sizes and environmental surroundings. Morris (1994) examined the performance of GeoSat over the Great Lakes (Erie, Huron, Michigan, Ontario and Superior), and the root mean square error (*RMSE*) between the altimetry-derived water
level estimates and the gauge measurements ranged from 9.4 to 13.8 cm. Birkett (1995) assessed Topex/Poseidon over Lake Ontario, Michigan and Superior, and its RMSE ranged from 4.69 to 6.2 cm. Also, Birkett *et al*. (2010) evaluated Jason-2 water level estimates against gauge measurements over five lakes. They found that its *RMSE* was 2.95 cm for Lake Ontario (with an area of ~20,000 km$^2$) and 33.2 cm for Lake Yellowstone (with an area of ~350 km$^2$). Frappart *et al*. (2006) investigated the performance of the four retrackers of ENVISat over three small lakes (with area from 100 to 300 km$^2$) near
Curuai in Amazon basin. They observed that the *Ice1 retracker* was the best for retrieving lake water levels with ENVISat altimetry observations. Jarihani *et al*. (2013) compared five different satellite radar altimetry missions (ENVISat, GFO, T/P, Jason-1 and Jason-2) and assessed the performance of different retrackers adopted by these missions over Lake Eildon (138 km$^2$) and Lake Argyle (1000 km$^2$) in Australia. They found out that among the five missions Jason-2 gave the best results with a *RMSE* of 28 cm for Ice1 retracker and 32 cm for MLE3 retracker, while T/P yielded the largest *RMSE* of 150 cm for
its sole Ocean retracker. Schwatke *et al.* (2015a) evaluated the performance of ENVISat and SARAL over the Great Lakes





and found that both missions can achieve very low *RMSE,* ranging from 2 – 6 cm for these large lakes. Villadsen *et al.* (2016) reprocessed Cryosat-2 data with several self-developed retrackers and assessed their performance over Lake Vanern ($5550 \ km^2$) and Lake Okeechobee ($1436 \ km^2$). They demonstrated that the Multiple Waveform Persistent Peak (MWaPP) retracker produced the lowest *RMSE* of 9.1 cm over Lake Vanern and 13.4 cm over Lake Okeechobee. Cretaux *et al.* (2018)

evaluated Sentinel-3 and Jason-3 over Lake Issykkul ( $6236 \ km^2$), and found that both missions achieved a very low *RMSE* of 3 cm with the Ocean retracker. Shu *et al.* (2020) assessed the performance of the Sentinel-3 SAR retrackers over fifteen lakes, and they reported that the SAR Altimetry Mode Studies and Applications -3 (SAMOSA-3) retracker has the lowest mean *RMSE* of 8.08 cm.

Apparently, each of those previous evaluations only focused on a few of radar altimetry missions. Those individual

evaluations are not strictly comparable, since each study was conducted over a different set of lakes. The differences in lake's size, geographic location, surrounding topography and land cover types could significantly influence the accuracy of lake water levels retrieved by satellite radar altimeters (Maillard et al., 2015).

Despite the previous research efforts, many questions remain as to the construction of a long-term time series of water level for ungauged inland lakes, particularly for those locate in remote areas (e.g. the Arctic coastal plains). As described above,

each radar altimetry mission spans different time periods and has different levels of measurement accuracy, and there exist systematical differences (biases) between different mission measurements. The question is: to construct a long-term consistent time series of lake water level estimates, which radar altimetry mission can be used as a high-confidence initial reference to remove the biases between missions and to tie different missions together? For a certain time period, one lake may be visited by multiple radar missions. In this case, which satellite radar altimetry mission may provide more reliable

lake water level estimates? Most of radar altimetry missions have several retrackers that can be used to estimate lake water level. For a given radar altimetry mission, which retracker is most reliable and accurate for lake water level retrieval? The pursuit of answers to these questions entails a comprehensive and consistent evaluation of all radar altimetry missions over the same set of lakes.

In this study, we will examine the performance of all historical and currently operational satellite radar altimetry missions,

except for HY-2A and Cryosat-2 missions. HY-2A was excluded from this study because of the difficulty in obtaining its data product. The exclusion of Cryosat-2 was due to its long repeat cycle orbit that does not allow the production of frequent co-located observations for evaluation. Water level estimates retrieved by different retrackers of the eleven radar altimetry missions will be assessed by using the corresponding gauge measurements on twelve lakes of various sizes distributed in four countries. After this introductory section, we will briefly describe these lakes and the gauge measurements in Section 2.

In Section 3, we will introduce the data sets collected by the eleven satellite radar altimetry missions and the different retrackers adopted by each mission. Then, we present the methods for processing the satellite radar altimetry data to determine lake water levels in Section 4. Next, we evaluate each altimetry mission and its retrackers in comparison with the *gauge* measurements in Section 5 and discuss the performance of each mission and relevant issues in integrating different





radar altimetry missions to construct consistent long-term time series in Section 6. The research findings are summarized in Section 7.

## 2 Case study lakes and gauge data

### 2.1 Case study lakes

Our case study sites include twelve lakes/reservoirs in four countries (as shown in Fig. 1). These lakes were selected based on the considerations that they were all overpassed by the eleven satellite radar altimetry missions and they are representative for different sizes, environmental surroundings, and winter ice conditions. The geographic location, the winter ice condition and the gauge station for these lakes are summarized in Table 1. The largest one is Lake Superior in North America (over 80,000 km$^2$), while the smallest one is Reservoir Lokka in Finland (about 500 km$^2$). The three lakes in Finland (Inarijarvi, Lokka, and Oulujarvi) and Lake Cedar in Canada all have numerous islands scattered within the lake, fragmenting the water surfaces of these lakes. Therefore, the surface condition of these lakes is very similar to small lakes, over which the satellite radar altimetry signal is contaminated easily by the surrounding land surfaces. These lakes are treated as small lakes to evaluate the performance of each satellite altimetry mission in contrast to the large lakes (e.g. the Great Lakes, Great Slave Lake, Lake Vanern). The boundary polygons of these twelve lakes were obtained from the Global Lakes and Wetland Databases (GLWD) (Lehner and Döll, 2004). The lake polygons were then used to extract measurements from each mission in the subsequent analysis.

These case study lakes have varying ice cover conditions in winter seasons, due to the differences in their latitudes and local climates. Among these lakes, Lake Inarijarvi in Finland is the northernmost with a latitude of 69.02° and Lake Erie is the southernmost with a latitude of 42.16°. The three lakes in Finland and the three lakes in Canada are fully ice-covered in winter seasons. The ice cover usually lasts more than 7 months for Lake Inarijarvi (Korhonen, 2006) and more than 5 months for Great Slave Lake (Howell et al., 2009). The duration of ice cover decreases for the lakes at more southern locations. In comparison with Canadian lakes, the ice cover on Finnish lakes is often much thinner (Shu et al., 2020) due to the heating effect of the North Atlantic Current (Rahmstorf, 2006;Korhonen, 2019).

Lake Vanern in Sweden and the Great Lakes of North America could be fully covered, partly covered, or totally free from ice in winter seasons depending on the winter air temperature. Lake Vanern often remains completely ice-free in winter. From 1979 to 2002, it was only covered by ice in nine winters (Weyhenmeyer et al., 2008). In a cold winter, Lake Superior and Lake Erie are often fully covered by ice, and the other three (Huron, Ontario, and Michigan) of Great Lakes are partly covered (Assel and Wang, 2017). While in warmer winters, all of them are partly covered.

### 2.2 Gauge data

*In-situ* water level measurements for the twelve lakes were collected respectively at the gauge stations listed in Table 1, which are obtained from four online databases. Those include the Finnish Environment Information Management System –





Hertta operated by Finnish Environment Institute (SYKE) (http://www.syke.fi/fi-FI/Avoin_tieto/Ymparistotietojarjestelmat), the SMHI (Swedish Meteorological and Hydrological Institute) website (http://vattenwebb.smhi.se/station/), the Canada Real-time Hydrometric Data Website (https://wateroffice.ec.gc.ca/ mainmenu/real_time_data_index_e.html), and the Center for Operational Oceanographic Products and Services (https://tidesandcurrents.noaa.gov/) operated by NOAA. These gauge stations measure the water-equivalent lake levels when the lake is ice-covered (Shu et al., 2020). Note that the gauge data are

referenced to different datums. In this study, only the gauge data on the Great Lakes are converted to EGM2008 using the tool VDatum (https://vdatum.noaa.gov/).

### 3 Satellite radar altimetry data products

In this study, we evaluate the performance of radar altimeters onboard eleven satellite missions. Those include all historical and currently operational satellite radar altimetry missions except for HY-2A and Cryosat-2. No data are available from HY-

2A mission launched by China. Cryosat-2 operates on a long-term repeat orbit (369 days) in order to obtain spatially dense coverage in polar regions, and it is difficult to form time series of co-located water level observations for inland lakes. Most of the altimetry data products of the eleven satellite radar altimetry missions have gone through several rounds of updating and refinements. We used the most up-to-date version of data product of each mission for the evaluation. The geographical coverage, operational time period, repeat cycle, footprint size and retrackers of these radar altimetry missions are

summarized in Table 2. The temporal coverage and the overlapping time periods of the eleven missions are illustrated in Fig. 2.

Satellite radar altimeters measure elevation through transmitting radar signal pulses to the nadir surface and timing the echoes. The transmitted and echoed radar pulse is sampled as pulse strength over the elapsed time, which is known as radar altimetry "waveform". Most of the eleven missions (except for GeoSat, Topex/Poseiden and GFO) adopted two or more

retracking algorithms (retrackers) to process the echoed waveforms in order to produce accurate elevation measurements for different types of Earth's surfaces. These retrackers can be divided into two general categories: the model-free retrackers and the model-based retrackers. The model-based retrackers fit a physically based model to the echoed waveform to produce elevation measurements. For example, the ENVISat Ocean retracker is based on the Brown model (1977) and the Sentinel-3 Ice-Sheet retrackers is based on a 5-part piecewise analytical function (MSSL/UCL/CLS, 2019). The model-free retrackers

have no assumption on the model of the echoed waveform, and the examples include Offset Center of Gravity (OCOG, also known as Ice1 or Ice) developed by Wingham (1986) and the Sea-Ice retracker developed by Laxon (1994).

Ten of the eleven missions (except for Sentinel-3) utilize the conventional pulse-limited altimeter to measure surface elevation. The diameter of the radar pulse footprint on Earth's surface varies from 1.6 km to 13.4 km, according to the satellite orbit, the echoing surface roughness and the duration of radar pulse (Chelton et al., 1989). Among the ten

conventional pulse-limited altimetry missions, SARAL utilizes a Ka-band (35.75 GHz) as the primary band with a bandwidth of 480 MHz to measure Earth's surface elevation, while the others use Ku-band (e.g. 13.6 GHz) as the primary





band with a bandwidth of 320 MHz. Due to the adoption of the Ka band and the higher bandwidth, the footprint generated by SARAL is about 0.8 times smaller than the other Ku-band altimeters for a given pulse length and orbit altitude (Raney and Phalippou, 2011). Sentinel-3 uses a synthetic aperture radar (SAR) altimeter to measure Earth's surface elevation. This SAR

altimetry technology decreases the along-track footprint size from several kilometers to about 300 m, which improves the retrieval of elevation information over more variable surfaces, e.g. coastal areas (Donlon et al., 2012).

*GeoSat* was launched on March 12, 1985 by U.S. Navy, and its operations consisted of two distinct mission phases: the Geodetic Mission (GM) and the Exact Repeat Mission (ERM) (McConathy and Kilgus, 1987). The GM phase lasted about 18 months from March 31, 1985 to September 30, 1986 and the ERM phase lasted about 3.5 years from November 8, 1986

to January 1990. In the GM phase, the satellite operated on a geodetic drifting orbit, while in the ERM phase, it operated on an exact-repeat orbit with a repeat cycle of 17 days. In both phases, the satellite collected elevation measurements of Earth's surface between 72°N and 72°S latitudes. GeoSat used a single ocean retracker based on the Brown (1977) model to produce elevation measurements for all different types of Earth's surface (Lillibridge et al., 2006). The georeferenced measurements were originally provided at a 1 Hz rate by the National Centers for Environmental Information (NCEI) at NOAA

(https://accession.nodc.noaa.gov/0053056). For this study, we obtained GeoSat data from the Radar Altimeter Database System (RADS) (Scharroo et al., 2013). RADS provides the most up-to-date harmonized geophysical and systematic corrections for all the satellite radar altimeters. The limitation of RADS is that all the data are provided only at 1 Hz rate. Since the original georeferenced data was also at the 1 Hz rate, the RADS GeoSat data product, instead of the NOAA/NCEI product, was therefore chosen for the evaluation. At the 1 Hz data rate, the sampling interval along the satellite track is 6 – 7

km depending on the latitude. *GeoSat Follow-On* (GFO) was launched on February 10, 1998 and ended on October 22, 2008. Since it was a follow-on mission of GeoSat, it retained the GeoSat ERM orbit with a repeat cycle of 17 days and covered Earth's surface between 72°N and 72°S latitudes along the satellite ground tracks (Office and Altimetry, 2002). The elevation measurements were produced by the same retracking algorithm used for GeoSat. The georeferenced data was provided at a 10 Hz rate and distributed by U.S. Navy and NOAA at https://accession.nodc.noaa.gov/0085960. With the 10

Hz sampling rate, the distance between two adjacent measurements is about 700 m.

*ERS-1* and *ERS-2* were launched by ESA on July 17, 1991 and April 21, 1995, and retired on March 10, 2000 and September 5, 2011, respectively (Duchossois and Martin, 1995). ERS-2 was the tandem mission of ERS-1 and carried basically the same set of instruments onboard ERS-1. ERS-1 had eight mission phases (Phase A, B, R, C, D, E, F, and G) with different repeat cycles during its lifetime (http://www.deos.tudelft.nl/ers/phases), including the 3-day cycle for the commissioning and

the ice phases (Phase A, B, and D), the 35-day cycle for the nominal observation phase (Phase R, C and G), and the 168-day cycle for the geodetic drifting phases (Phase E and F). ERS-2 had two phases: the 35-day nominal observation phase (from April 29, 1995 to February 21, 2011) and the 3-day phase (from March 10, 2011 to July 6, 2011). Elevation measurements collected by both missions cover Earth's surface between 81.5°N and 81.5°S latitude (Brockley, 2014). After the retirement of ERS-2, the data collected by the two missions between August 1991 and July 2003 were reprocessed to generate an

improved homogeneous long-term dataset, which is called the REAPER (the Reprocessing of Altimeter Products for ERS)





products (Brockley et al., 2017). In the reprocessing, the four retrackers used for ENVISat (*Ocean, Ice1, Ice2* and *Sea-Ice*) were adopted to refine elevation measurements. *Ice1* and *Sea-Ice* are model-free retrackers developed by Wingham (1986) and Laxon *et al.* (1994). The other two are model-based retrackers. Later, the ERS-2 data were further reprocessed by the Centre de Topographie des Océans et de l'Hydrosphère (CTOH) at the Laboratoire d'Etudes en Géophysique

etOcéanographie Spatiales (LEGOS) (Frappart et al., 2016). The CTOH ERS-2 product contains elevation measurements generated by two retrackers: *Ice1* and *Ice2*. In this study, we chose the ERS-1 REAPER data product from ESA (https://earth.esa.int/) and the further improved ERS-2 data product from CTOH (http://ctoh.legos.obs-mip.fr/) for the evaluation. Both products provide georeferenced elevation measurements at a 20 Hz rate. At this data rate, the distance between two adjacent measurements along the satellite track is about 350 m.

*ENVISat* was launched on February 28, 2002, as the successor to ERS-1 and ERS-2. In the nominal observation phase, ENVISat operated on the same orbit as ERS-1 and ERS-2 with a 35-day repeat cycle from 2002 to 2010. In October 2010, it was maneuvered to a new orbit with a repeat cycle of 30 days to extend its mission lifetime, until April 08, 2012. This new phase is referred to as "Extension Phase". In both phases, the elevation measurements were provided at an 18 Hz rate with a sampling interval of about 370 m along the satellite ground track. ENVISat mission used four retrackers (Ocean, Ice1, Ice2,

and Sea-Ice) to generate elevation measurements for different types of Earth's surface. In 2018, the ENVISat altimetry data were reprocessed and released by ESA as the ENVISat V3 product. We obtained this most recent Version 3 product from ESA (https://earth.esa.int/) for the evaluation.

*SARAL* is a joint altimetry mission of CNES (Space Agency of France) and ISRO (Indian Space Research Organization). It was launched on February 25, 2013 by ISRO and is the first satellite mission with a Ka-band (35.75 GHz) radar altimeter

onboard (Frappart et al., 2015;Bonnefond et al., 2018). During its exact repetitive phase from the launch to July 4, 2016, SARAL flew on ENVISat nominal orbit with a 35-day exact repeat cycle. Due to technical issues with the reaction wheels, the repetitive orbit was no longer maintained since July 4, 2016 and the orbit of the satellite decayed naturally, leading to irregular drifting ground tracks on Earth's surface. This new phase is known as "SARAL Drifting Phase" (Dibarboure et al., 2018). The four ENVISat retrackers (Ice1, Ice2, Sea-Ice and Ocean) were adopted by SARAL in the creation of different

data products for different types of Earth's surfaces. The data are provided at a rate of 40 Hz by AVISO+ (Archiving, Validation and Interpretation of Satellite Oceanographic data, ftp://avisoftp.cnes.fr/AVISO/pub/) at the CNES (https://aviso-data-center.cnes.fr/). The distance between two adjacent measurements along the satellite track is about 180 m. In this study, we only evaluated the SARAL data collected in the exactly repetitive phase.

*Topex/Poseidon (T/P)*, *Jason-1*, *Jason-2* and *Jason-3* are four continuous missions that provide long-term consistent

altimetry observations of Earth's surface along the same fixed ground tracks. The operation of each satellite is usually composed of two phases: the phase with nominal orbit and the phase with interleaved orbit (Fu et al., 1994). Both orbits have an exact repeat cycle of 10 days and cover Earth's surface between 66°N – 66°S latitudes. Each satellite in this series firstly flies on the nominal orbit after launch, and was usually maneuvered to a new orbit, a number of months after the launch of its successor satellite. The ground tracks generated by this new orbit phase are on the midway between its nominal ground





tracks, hence the new orbit is referred to as interleaved orbit. The period between the launch of the successor satellite and the maneuver of the predecessor satellite is often called tandem phase. During this phase, the two satellites fly on the same orbit separated by 60 – 70 seconds (see Jason-3 Products Handbook). Topex/Poseiden was launched on August 10, 1992 and then maneuvered to the interleaved orbit on August 15, 2002 after the launch of Jason-1 on December 7, 2001. Topex/Poseiden was decommissioned on October 9, 2005. The Topex/Poseiden data products were generated with its sole Brown-model

based retracker (herein after referred to as the Ocean retracker) (Rodríguez and Martin, 1994) for all different types of surfaces. In the original Topex/Poseiden data products, the geographic coordinates were provided for the 1 Hz elevation measurements. In this study, we utilized the data products created by RADS for the evaluation. The distance between two adjacent 1-Hz measurements along satellite track is about 6 km. Jason-1 was shifted to the interleaved orbit on February 10, 2009, after the launch of Jason-2 on June 20, 2008. Jason-1 stayed on the interleaved orbit for 3 years until May 7, 2012

when it was adjusted to a geodetic orbit. It was finally decommissioned on July 1, 2013. Jason-2 was transferred to the interleaved orbit on October 17, 2016, after the launch of Jason-3 on January 17, 2016. It maintained the interleaved orbit for 8 months and then transferred to a geodetic orbit on July 10, 2017. It was decommissioned on October 1, 2019. Jason-3 now operates on the nominal orbit and will continue until the planned launch of Jason-CS/Sentinel-6 in 2020. Two retrackers have been used by all three missions to generate elevation measurements: the Brown-model based MLE4 retracker for ocean

surfaces and the model-free Ice retracker (similar to OCOG/Ice1 retracker) for non-ocean surfaces (see the Jason-1, 2 &3 Products Handbook for details). Another Brown-model based retracker MLE3 has also been adopted for Jason-2 and Jason-3. Due to its apparent inferior performance in comparison with MLE4 (Thibaut et al., 2010;Vu et al., 2018), it is not included for our evaluation. All these three radar altimetry missions provide elevation measurements at a rate of 20 Hz. The ground distance between two adjacent measurements is about 350 m. We obtained the altimetry data products of these three

missions from AVISO+ for the evaluation.

The *Sentinel-3* mission consists of two identical satellites, the Sentinel-3A and Sentinel-3B, which were launched on February 16, 2016 and April 25, 2018, respectively. The ground tracks of Sentinel-3B fall exactly in the middle of the ground tracks of Sentinel-3A. In other words, the Sentinel-3B is operated on an interleaved orbit, in parallel with the Sentinel-3A on the nominal orbit. The two orbits have the same 27 days repeat cycle and collect elevation measurements

along their ground tracks between 81.35°N and 81.35°S latitudes (Donlon et al., 2012). Both satellites carry a Synthetic Aperture Radar altimeter instrument (SRAL) for the elevation measurements. The SRAL works primarily on the Synthetic Aperture Radar (SAR) mode with the Low Resolution Mode (LRM) as a back-up (Sentinel-3-Team, 2017). Four retrackers are used in the SAR mode to produce elevation measurements, including SAR Altimetry Mode Studies and Applications -3 (SAMOSA-3), Offset Center of Gravity (OCOG), Sea-Ice, and Ice-Sheet (Shu et al., 2020). The OCOG (also known as Ice1)

is a model-free retracker developed by Wingham (1986). The other three are model-based fully-analytic or semi-analytic retrackers (Shu et al., 2020). Due to the high rate of missing data (Shu et al., 2020), the Sea-Ice retracker is not included for the evaluation in this study. The elevation measurements are provided at a rate of 20 Hz. The interval between two adjacent





measurements along the satellite track is about 300 m (Sentinel-3-Team, 2017). We obtained the Sentinel-3 altimetry data from the ESA Copernicus Open Access Hub (https://scihub.copernicus.eu/) for the evaluation.

In this study, the altimetry data collected by each mission in geodetic phase (or drifting phase) are not included in the evaluation. In the geodetic phase, the drifting ground tracks do not generate frequent observations for a specific lake to form time series of water level measurements. In this study, for all the completed missions, only the data collected in their exact repeat phase are used for the evaluation. For instance, the data collected in Phase ERM were used for GeoSat and the data collected in Phase R, C and G were used for ERS-1. In the "Extension Phase" of ENVISat mission and in the intermittent

phases of Topex/Poseiden, Jason-1 and Jason-2 missions, the satellites all operated on exact repeat orbit. Therefore, the data collected in these phases were also included in the evaluation. For the two currently operational missions Sentinel-3 and Jason-3, the observations for longer than a full year (including winter and summer) are used for the evaluation, namely, Jason-3 data between February 2016 and March 2018 and Sentinel-3 data between June 2016 and September 2017.

In addition to the altimeter instrument, most of the eleven satellite missions (except for GeoSat) also carried a passive

microwave radiometer (MWR) to simultaneously measure the brightness temperature (referred to as $T_B$) of Earth's surface. The microwave bands adopted by each mission are listed in Table 2.

## 4 Lake water level determination and accuracy evaluation methods

The method used to determine lake water level from satellite radar altimetry in this study consists of three technical data processing steps. First, the surface elevation measurements are retrieved from altimetry data products of the eleven satellite

missions for the twelve case study lakes, and the most recent release    e of the altimetry data products with the up-to-date geophysical corrections have been used. Second, the spurious surface elevation measurements are filtered out through statistical analysis, and the remaining valid surface elevation measurements within a lake are statistically aggregated to determine lake water level at different time points. Third, the ice-cover condition is examined using the simultaneous $T_B$ measurements from the MWR instruments, and those lake water level estimates during the ice-covered period are excluded

in the subsequent accuracy evaluations. To evaluate the performance of each satellite altimeter and its retrackers, three accuracy measures, including the Pearson's correlation coefficient $r$, the bias and the $RMSE$, have been calculated by comparing the radar altimetry derived lake water level estimates with the corresponding *gauge* measurements.

### 4.1 Retrieval of lake surface elevation measurements

Following Crétaux *et al.* (2017), the surface elevation is determined for each satellite radar altimetry mission according to

Equation 1:

$$h_{retrk} = H - R_{retrk} - (\Delta R_{iono} + \Delta R_{wet} + \Delta R_{dry} + \Delta R_{solidEarth} + \Delta R_{pole}) - Geoid \qquad (1)$$

where $h_{retrk}$ is the surface elevation generated by a retracker, $H$ is the height of satellite orbit, $R_{retrk}$ is the range between the satellite and the nadir Earth's surface generated by a retracker, $\Delta R_{iono}$, $\Delta R_{wet}$ and $\Delta R_{dry}$ compensate the delay of radar





pulse due to the ionosphere, the wet troposphere and the dry troposphere, respectively, $\Delta R_{solidEarth}$ and $\Delta R_{pole}$ are for solid

Earth tide correction and pole tide correction, $Geoid$ converts the reference surface from ellipsoid to geoid. In this study, the geoid model EGM2008 is adopted.

Due to the variable nature of Earth's atmosphere, the three atmospheric components ($\Delta R_{iono}$, $\Delta R_{wet}$ and $\Delta R_{dry}$) have significant influences on the accuracy of altimetry measurements (Fernandes et al., 2014;Fernandes and Lázaro, 2016;Crétaux et al., 2009;Scharroo and Smith, 2010). Many global atmospheric models have been used to quantify the

biases induced by the three atmospheric components at different locations and times. For the ionospheric correction ($\Delta R_{iono}$), it has been recommended to use the NIC09 (New Ionosphere Climatology) model for the radar altimetry measurements acquired before September 1998 (Scharroo and Smith, 2010) and to use the GIM (Global Ionosphere Map) model for the measurements acquired after that time (Iijima et al., 1999). For the dry and the wet tropospheric corrections ($\Delta R_{dry}$ and $\Delta R_{wet}$), the three most commonly used atmospheric models are produced by the European Centre for Medium-

Range Weather Forecasts (ECMWF) and the National Centers for Environmental Prediction (NCEP). Those include the ECMWF model (Miller et al., 2010), the ECMWF Re-Analysis Interim (ERA) model (Dee et al., 2011), and the NCEP model (Caplan et al., 1997). The magnitude of the dry and the wet tropospheric corrections depends linearly on the height of the surface over which the altimetry measurement is made. The higher the surface elevation, the smaller the magnitude of the dry and the wet tropospheric correction terms. The difference between the dry tropospheric corrections computed at the sea

surface with an elevation of 0 m and at the surface with an elevation of 5000 m could be as high as 1 m (Fernandes et al., 2014). For all the altimetry data products adopted in this study, the dry and the wet tropospheric corrections were computed with the height of the surface where the altimetry measurements were taken. Table 3 lists the version of each altimetry data product and the models of the three atmospheric corrections utilized in this study.

**4.2 Statistical determination of lake water levels**

The twelve case lakes in this study were all overpassed by the eleven satellite radar altimetry missions. The number of each mission's ground tracks on these lakes is determined by the size of lake and the satellite orbit. The large lakes (Lake Superior) usually have multiple ground tracks for each mission, while the small lakes (e.g. Lokka) may have only one ground track for a satellite mission. For lakes with more than one ground tracks, we selected the one near the gauge station for the evaluation, as listed in Table 4. The total number of completed cycles for each mission depends on its operational

lifetime and the temporal length of a repeat cycle. For a mission with long lifetime and short repeat cycle, the overpass number could be much higher. As listed in Table 4, Topex/Poseiden has the highest number of complete cycles (333 in the nominal phase and 111 in the intermittent phase). In each repeat cycle, there is one satellite overpass along the selected ground track for each mission.

For each satellite overpass during the exact repeating phase, we first extracted the surface elevation measurements along a

ground track falling within lakes using lake polygons from the GLWD. Then, the extracted elevation measurements along



each ground track were combined to form a surface elevation profile, which was examined to filter out the spurious measurements with the robust Median-Absolute-Deviation (*MAD*) statistic (Shu et al., 2018;Liu et al., 2012;Shu et al., 2020). The spurious measurements could possibly be induced by the contamination of land surface when the satellite ground track passes through lake islands or when it is close to lake shore. The median of the remaining elevation measurements
along the track is then used as the estimate of the lake water level on the day of each satellite overpass. Finally, the time series of water level estimates were evaluated through comparing with the concurrent gauge measurements.

### 4.3 Identification of lake level estimates affected by ice cover

It has been demonstrated that the lake ice cover in winter could have strong influences on the radar altimetry signal pulse, resulting in lower elevation measurements than the real lake surface elevation (Birkett and Beckley, 2010;Ziyad et al., 2020).
The mechanism on how lake ice deforms the radar altimetry signal pulse and fails the waveform retracking algorithms has been investigated in (Shu et al., 2020). In this study, we followed the method in (Shu et al., 2020) to examine the ice cover condition for all satellite radar altimetry missions over the case study lakes.  Namely, we examine the temporal variations of brightness temperature ($T_B$) over lake surface to detect the lake ice cover. Similar to the preprocessing of radar altimetry surface elevation measurements, we first filter the simultaneous microwave $T_B$ measurement profile along the track over a
lake. Then, all the remaining valid microwave $T_B$ measurements were averaged to represent the temperature for the day of each satellite overpass. The time series curve of $T_B$ was then analyzed to determine the dates of ice-on and ice-off for each winter, indicated by the sudden increase and rapid decrease of $T_B$ on the curve. Those radar altimetry measurements collected in the ice-covered condition were identified and then excluded from the subsequent evaluations.

### 4.4 Accuracy measures for the performance evaluation

The performance of a satellite altimetry mission and its retrackers were evaluated in terms of three accuracy measures as in (Shu et al., 2020), including the Pearson's correlation coefficient (*r*), the bias (*Bias*) and the root mean square error (*RMSE*). The *Bias* and the *RMSE* were computed as below.

$$Bias = \frac{1}{n}\sum_{i=0}^{n}(H_{retrk}^{i} - H_{gauge}^{i}) \tag{2}$$

$$RMSE = \sqrt{\frac{1}{n}\sum_{i=0}^{n}(H_{retrk}^{i} - H_{gauge}^{i} - Bias)^2} \tag{3}$$

where *n* is the total number of a satellite mission's overpasses along the selected track on a lake, *i* is the index of an overpass, $H_{retrk}^{i}$ is the altimetry-derived lake level estimate for satellite overpass *i* given by a specific retracker, $H_{gauge}^{i}$ is the concurrent gauge measurement at the time of overpass *i*.

These three accuracy measures are computed for each retracker of each mission over each lake. The *Bias* represents the systematic (positive or negative) difference between the series of altimetry-derived estimates and the gauge measurements. If
both are referenced to the same vertical datum (e.g. EGM2008), then the smaller the *Bias*, the closer altimetry-derived





estimates to the real lake water level. Since the datums of the altimetry-derived water levels and the gauge measurements were consistent only for the Great Lakes as mentioned in Section 2.2, we compared and evaluated the biases of all the retrackers of the eleven missions for these five lakes. The $r$ indicates each retracker's capability in depicting lake water level temporal variation. A high $r$ value shows that the retracker captures the lake water level variation very well. Note that the

correlation coefficient $r$ is not affected by systematic errors/biases or vertical datum differences. In our evaluation, the *RMSE* is calculated after the *Bias* of each retracker over each lake was removed (Shu et al., 2020). The *RMSE*, hence, represents the relative accuracy (precision) of the altimetry-derived lake level estimates. By removing the *Bias*, the inconsistency between the vertical datums of the altimetry-derived water levels and the *gauge* measurements would not affect RMSE values, making all the retrackers over the twelve lakes comparable to each other in terms of RMSE value.

## 5 Results

### 5.1 Radar altimetry derived Lake water level estimates

Fig. 3 shows the time series of $T_B$ and altimetry-derived water levels over Great Slave Lake collected by ENVISat, Jason-2 and Sentinel-3 in the winters of 2003/2004, 2011/2012, and 2016/2017, respectively. The ice-covered duration is determined by the sudden increase and the decrease of $T_B$, as indicated by the vertical dash lines in Fig. 3a, 3b and 3c. The similar

temporal variation of $T_B$ was also observed for other satellite missions over other lakes when they were covered by ice. As shown in Fig. 3d, 3e and 3f, the lake water level estimates during the ice-covered periods deviate significantly from the gauge measurements, while during the ice-free seasons the lake water level estimates correlate very well with the gauge measurements.

Table 5 summarizes the number of lake level estimates during ice-free (open water) and ice-covered seasons over each lake

for each retracker of the eleven missions. For some satellite missions, the number of valid lake water level estimates over a certain lake during ice-free season was too small to perform an evaluation. For example, the number of GeoSat estimates over Lake Inarijarvi, Lake Lokka, Lake Oulujarvi and Lake Cedar are all less than three. Therefore, the evaluation of GeoSat over these lakes was not conducted. As shown in Table 5, the total number of lake water level estimates ( sum of the ice-covered number and the ice-free number) for some satellite missions, such as GeoSat and GFO, are considerably smaller

than the number of completed orbit cycles, due to satellite data loss. The reasons for satellite data loss could be the malfunction of the sensor, the maneuver of the satellite during the phase transition, the failure of the retracker to reach convergence when processing complex waveforms (e.g., multi peaks) from inhomogeneous reflecting surfaces in the altimeter footprint, saturation of the sensor over very bright targets, or the rapid changes of the topography that are larger than the size of tracking window causing tracking losses (Biancamaria et al., 2017).

We calculated the data loss rate of lake level estimates over each lake for each retracker of the eleven missions. As shown in Table 6, GeoSat has very high data loss rate for almost all the lakes. The average data loss rate is 65.42%. There are seven lakes with a loss rate higher than 70%. Particularly, the data loss rate over small lakes is much higher than that for large





lakes. The highest data loss rate is 98.51% over Lake Cedar. Similarly, GFO also has very high data loss rate for small lakes. The highest data loss rate is 80% over Lake Cedar. The high data loss rates of GeoSat and GFO hamper their usefulness for

retrieving lake water levels. In contrast, SARAL and Sentinel-3 have a very low data loss rate over both large lakes and small lakes. For ERS-2 and ENVISat, the data loss rates over small lakes are slightly higher than that over large lakes. Another interesting observation is that on average the model-based retrackers have a relatively higher data loss rate than model-free retrackers for all missions. For example, the data loss rates of MLE4 retracker of Jason-1, -2 and -3 missions are 21.48%, 16.45% and 15.7%, about twice as high as the loss rates of the Ice retracker of these three missions (12.8%, 6.61%

and 6.20%). It suggests that the model-free retrackers are more reliable than model-based retrackers for producing continuous lake water level estimates, confirming the observations of Frappart *et al*. (2006) and Sulistioadi *et al.* (2015).

### 5.2 The *Biases* of altimetry-derived lake water level estimates

We construct a long-term series of lake water levels for each of the twelve lakes using the altimetry-derived estimates during ice-free seasons. Fig. 4a shows the lake water level time series over Great Slave Lake. For many satellite missions, there are

more than one water level time series from different retrackers. Fig. 4a displays only the water level time series produced by the retracker of each mission that has the lowest data loss rate in Table 6. For example, the water level time series produced by Jason-1 Ice retracker, rather than MLE4 retracker, were displayed.

Clearly, *biases* exist between the altimetry-derived estimates and the gauge measurements for all missions. The magnitude of the *biases* varies among the missions. The small magnitude of bias indicates that the absolute values of altimetry-derived

lake water level are close to the ground truth represented by gauge measurements. As shown in Fig. 4a, the time series of T/P water level estimates (given by Ocean retracker) have the least difference to the gauge data on Great Slave Lake in absolute values, while the time series of ERS-2 estimates (produced by Ice1 retracker) have the largest absolute difference from the gauge measurements. As shown in Fig. 4b, after removing the *biases*, the altimetry-derived estimates match the gauge measurements well for most of the missions over Great Slave Lake.

The *bias* value for each retracker of the eleven missions over the twelve lakes are reported in Table 7. Since only the Great Lakes' gauge measurements are referenced to the same vertical datum as altimetry-derived lake water levels, we will then focus our discussion of the *bias* on these five Great Lakes. For a specific lake (e.g. Lake Erie) the different missions and different retrackers of the same mission could have very different magnitudes of *biases*. The mean bias is calculated for each retracker by averaging the *biases* over the five Great Lakes. As shown in Table 7, the retrackers with a mean *bias* less than

10 cm include the Ocean retracker of Topex/Poseiden mission, the MLE4 retracker of Jason-1, -2 and -3 missions, and the Ice-Sheet and SAMOSA3 retracker of Sentinel-3 mission. The mean *Bias* of Jason-3 MLE 4 retracker is less than 1 cm. Note that all those low bias retrackers are model-based. Actually, for all missions with multiple retrackers, the model-based retrackers outperforms the model-free retracker in terms of mean *bias* over the Great Lakes.





### 5.3 The performance of radar altimetry missions in capturing lake water level dynamics

The Pearson's correlation coefficient $r$ was calculated for all the retrackers of each mission over every lake that has more than three lake water level estimates. A high correlation coefficient of the lake water level estimates from a retracker with *gauge* measurements indicates a strong capability of the retracker in reconstructing the temporal variation of lake water levels. As shown in Table 8, all the retrackers of the eleven missions, except for ERS-1 Sea-Ice retracker, have a good performance on large lakes (e.g. the Great Lakes). In contrast, many retrackers give a $r$ value less than 0.7 over small lakes.

ERS-1 Ocean retracker gives the lowest $r$ value of 0.07 over Lake Oulujarvi.

The performances of SARAL and Sentinel-3 missions in capturing the lake water level dynamics are outstanding. Almost all their retrackers produce a very high $r$ value over both large and small lakes. Their stronger capabilities than other satellite radar missions of retrieving water levels for small waterbodies were previously reported in (Bogning et al., 2018;Normandin et al., 2018). The Sentinel-3 Ice1 retracker gives the highest mean $r$ value (0.96) across the twelve lakes. In contrast, ERS-1

Sea-Ice retracker has very poor performance over almost all the lakes, even on very large lakes, resulting in the lowest mean r value of 0.50.

As indicated in Table 8, for all the missions the model-free retrackers (except for ERS-1 Sea-Ice retracker) outperform the model-based retrackers in depicting water level variations over small lakes. The model-free retrackers, including the Ice1 (or OCOG) retracker of ERS-1, ERS-2, ENVISat, SARAL and Sentinel-3 missions, and the Ice retracker of Jason-1, -2 and -3

missions, all yield higher $r$ values than model-based retrackers of the same missions over small lakes. The performance contrast between model-free and model-based retrackers is particularly conspicuous over Lake Oulujarvi and Lake Vanern. Fig. 5 shows the scatterplots produced by the model-free retrackers of ERS-1, Jason-2 and Sentinel-3 over the lakes Oulujarvi, Vanern and Erie. Fig. 6 shows the corresponding scatterplots produced by the model-based retrackers (ERS-1 Ocean, Jason-2 MLE4 and Sentinel-3 SAMOSA3) of the same missions over the three lakes. Apparently, the estimates given

by model-free retrackers correlate very well with gauge measurements for all three missions over the three lakes. The correlation is higher on large lakes (e.g. Lake Erie) than on small lakes (e.g. Lake Oulujarvi). In contrast, no clear correlation can be observed between the water level estimates from ERS-1 Ocean retracker and Jason-2 MLE4 retracker and gauge measurements on Lake Oulujarvi. The correlation of Jason-2 MLE4 retracker estimates with gauge measurements on Lake Vanern is very low. It suggests that in comparison with the model-based retrackers, the model-free retrackers

(OCOG/Ice1/Ice) are less affected by the contamination of land surface surrounding small lakes.

### 5.4 Overall precision of altimetry-derived lake water level estimates from different missions

As introduced in Section 4.4, the RMSE was computed for each retracker after removing the bias, which contains the vertical datum difference between satellite and ground measurements and systematic error between the gauge station and retrackers. Such calculated RMSE represents the precision of altimetry-derived lake water level estimates as compared with gauge

measurements. A small RMSE of a retracker means a small random error, hence a high precision of the retracker in





retrieving lake water levels. The RMSE values for all retrackers of the eleven missions over the twelve lakes are listed in Table 9. Similar to the pattern that we observed for the correlation coefficient r, the RMSE values for large lakes are significantly smaller than those for small lakes. Most retrackers of the eleven missions have a RMSE less than 10 cm for large lakes. The RMSEs for small lakes, however, may exceed 30 cm. Among all retrackers and all missions, SARAL Ice2

retracker givers the lowest RMSE of (1.92 cm) over Lake Ontario, while GFO produces the highest RMSE of 132.81 cm over Lake Oulujarvi. Again, it reflects the adverse influences of land surface on the accuracy of satellite altimeters in the retrieval of lake water levels for small lakes.

As compared to other missions, Sentinel-3 and SARAL have clearly better measurement precision in terms of RMSE over small lakes, such as Lake Inarijarvi, Lokka and Oulujarvi, which are largely due to the smaller footprint of the altimeters

onboard these two missions. Most retrackers of these two missions yielded a RMSE less than 30 cm over the three lakes. In contrast, the RMSEs of ERS-1 retrackers over these three lakes are mostly higher than 50 cm. The mean RMSEs of the three Sentinel-3 retrackers (7.31 cm for Ice-Sheet, 6.08 cm for OCOG and 6.57 cm for SAMOSA3) are much smaller than other missions. The mean RMSEs of the SARAL retrackers (7.89 cm for Ice1, 7.30 cm for Ice2, 8.85 cm for Sea-Ice and 10.46 cm for Ocean retracker) are slightly higher than Sentinel-3 retrackers.

For the same mission, model-free retrackers often have lower RMSEs than the model-based retrackers. For example, the average RMSEs across the twelve lakes are 14.76 cm for ERS-1 Ice1, 11.28 cm for Jason-1 Ice, 7.74 cm for ENVISat Ice1, 8.18 cm for Jason-2 Ice and 8.03 cm for Jason-3 Ice retracker. In contrast, the average RMSEs are 35.17 cm for ERS-1 Ocean, 18.68 cm for Jason-1 MLE4, 14.66 cm for ENVISat Ocean, 19.22 cm for Jason-2 MLE4 and 17.15 cm for Jason-3 MLE4 retracker. The mean RMSE of the model-based retrackers is approximately twice as large as that of the model-free

retrackers. The performance contrast in terms of RMSE between the two types of retrackers is striking for small lakes. On Lake Oulujarvi, the RMSEs for the Ice retracker of Jason-1, -2 and -3 missions are 17.42 cm, 17.16 cm and 24.65 cm. But, the RMSEs of the MLE retracker of these three missions are 124.98 cm, 99.91 cm and 110.32 cm, 5 – 6 times higher than the model-free retrackers. Again, it highlights the fact that model-free retrackers are more precise choices for the retrieval of water levels for small lakes. For large lakes, both types of retrackers have similar performance in lake level estimates.

Therefore, the selection of either model-free or model-based retracker does not make much difference in the precision of water level estimates for large lakes.

## 6 Discussion

Among the eleven satellite radar altimetry missions, eight of them have more than one retrackers to measure the Earth's surface elevation. It should be noted that none of these retrackers were dedicated to the surface elevation measurements of

inland lakes. Our evaluation intention is to identify which retrackers have relatively better performance. As shown in Tables 6, 8 and 9, all the retrackers of the same mission have similarly good performance for large lakes (e.g. the Great Lakes) in terms of the data loss rate, the correlation coefficient *r* and *RMSE*. In other words, any of the retrackers for the same mission





(except for the ERS-1 Sea-Ice retracker) could be used to retrieve water levels for a large lake. However, for small lakes, the model-free retrackers, such as the Ice1 (OCOG) retracker of ERS-1, ERS-2, ENVISat, and SARAL, and the Ice retracker of

Jason-1, -2 and -3, are clearly better choices than the model-based retrackers, such as the Ocean retracker of ERS-1, ERS-2, ENVISat, and SARAL, and the MLE4 retracker of Jason-1, -2 and -3, or the non-model based retracker Sea-Ice. Our evaluation result is contrary to Sulistioadi (2015), who found comparable performances between Sea Ice and OCOG retrackers over a couple of small lakes using ENVISat data. In previous study, Frappart (2006) concluded that the model-free Ice1 retracker was the best among the four ENVISat retrackers in the retrieving lake water levels. Our evaluation results

consistently demonstrate that for all radar altimetry missions model-free retrackers tend to have high correlation coefficients, lower data loss rates and *RMSE*s than the model-based retrackers over small lakes. The model-free retrackers are therefore recommended for the retrieval of water levels over small lakes.

It is evident that the performance of the satellite radar altimetry missions has been improving with the time, as observed from Tables 6, 8 and 9. In general, the new generation of radar altimetry mission performs better than historical missions. The data

loss rate decreases from 65.42% for the first-generation mission of GeoSat to 2.32% for the currently operational Sentinel-3 mission. The mean *RMSE* decreases from 35.17 cm of the early ERS-1 mission to 6.47 cm of the current Sentinel-3 mission. Among the eleven missions, the most recent Sentinel-3 mission has the best performance. All three retrackers (particularly the OCOG retracker) produced the lowest mean *RMSE*s and the lowest mean data loss rate among all historical and currently operational missions. The reason is that the SAR altimeter onboard Sentinel-3 increases the along-track sampling resolution

(~300 m) and maximizes the information retrieval over variable terrain surfaces (Donlon et al., 2012).

Following Sentinel-3, SARAL gave the second-best performance among these missions. The Ice1 retracker of SARAL performed well for both small lakes and large lakes. For the period between February 2013 and June 2016, the SARAL Ice1 retracker provided the best retrieval of water levels for the overpassed lakes, due to its smaller footprint and larger bandwidth owing to the use of Ka-band (Bonnefond et al., 2018). Between February 2002 and April 2012, the Ice1 retracker of

ENVISat mission provided very accurate retrieval of lake levels. Overall, the ENVISat Ice1 retracker gave slightly better results than the Ice retracker of Jason-1 and Jason-2 missions. However, since the repeat cycle of ENVISat is 35 days and the data loss rate of ENVISat Ice1 retracker is almost twice as high as that of Jason-1 and Jason-2 missions, the two Jason missions (with a repeat cycle of 10 days) provided temporally more frequent and continuous estimates of lake water levels than ENVISat. It should be noted that Jason-1 and Jason-2 cover only the Earth's surface between 66°N and 66°S latitudes.

For lakes located at high latitude polar regions, ENVISat is the best alternative option during its operational time.

GFO has a much higher data loss rate than other contemporary missions. For the lakes overpassed by GFO, ERS-2 and Topex/Poseiden in the same period of time, GFO is the least desirable choice. For the period from 1991 to 2001, ERS-1 and ERS-2 are better choices for small lakes than Topex/Poseiden. But for large lakes, Topex/Poseiden should be adopted since it has much more frequent overpasses than ERS-1 and ERS-2 satellites, although comparable accuracy for lake level

estimates. GeoSat exhibited a good performance for large lakes (e.g. the Great Lakes). Even though it has an extremely high




data loss rate for almost all the twelve lakes, the water level estimates given by GeoSat are still valuable since it was sole satellite radar altimetry mission between 1985 and 1989.

To construct a long-term time series of lake water level for an ungauged lake, one critical step is to determine a reference mission to tie all satellite missions together by compensating the biases between them. A reference mission should meet two

requirements. First, the reference mission should be able to provide precise lake level estimates, at least comparable with other missions. Second, the reference mission should have a long operational time period so that it has temporal overlaps with many other missions. Both Sentinel-3 and SARAL meet the first requirement, due to their high performance for both large and small lakes. However, they have a relatively short temporal overlap with other missions and do not satisfy the second requirement. Among eleven radar altimetry missions, there are four missions that have a nominal operational time

over 10 years (the geodetic phase not counted), including Topex/Poseiden, Jason-1, ENVISat and Jason-2. Topex/Poseiden does not meet the first requirement well since its performance is apparently inferior to Jason-1, ENVISat and Jason-2 in terms of the $r$, $RMSE$ and the data loss rate. Despite its long data duration, ENVISat has higher data loss rate and longer repeat cycle, hence less frequent water level estimates than Jason-1 and Jason-2 missions, which reduces the chance of concurrent overpasses of ENVISat with other missions over the same lake. In comparison, Jason-2 is a better choice as the

reference mission than Jason-1. First, the Ice retracker of Jason-2 has much smaller $RMSE$ and lower data loss rate than Jason-1, as shown in Table 6 and 9. The Jason-2 Ice retracker's performance ($r = 0.93$, $RMSE = 8.18$ cm) in retrieving lake water levels is close to the best performance retracker-Sentinel-3 OCOG ($r = 0.96$, $RMSE = 6.47$ cm). Second, Jason-2 temporally overlapped with 7 other missions, including ERS-2, GFO, Jason-1, ENVISat, Saral, Jason-3 and Sentinel-3. Jason-1 has 6 overlaping missions as shown in Table 2. Third, Jason-2 has a short repeat cycle of 10 days, hence the better

chance to find concurrent overpasses with other missions over the same lake. Moreover, for the four Topex/Poseiden–Jason satellites, the predecessor and the successor satellites measure the same location almost the same time (separated by 60 – 70 seconds) during their tandem phases. This allows for the accurate estimation of the inter-mission biases between them over the large lakes around the world. For example, based on the measurements during the tandem phases over the five Great Lakes, the estimated biases (with the successor satellite as the benchmark) are 0.48±4.48 cm for Topex/Poseiden and Jason-

1, 19.56±5.38 cm for Jason-1 and Jason-2, and -20.47±0.16 cm for Jason-2 and Jason-3. Using Jason-2 as the initial reference, we are able to form a consistent Topex/Poseidon-Jason series of water level estimates that overlaps with all other radar altimetry missions (except for GeoSat). This consistent series of water level estimates can be further used as the reference for other missions to estimate the biases between them, then construct the long-term time series of water level records at global scale. As discussed above, the model-free retrackers outperform the model-based retrackers over small

lakes. For the purpose of constructing consistent long-term time series of lake water levels, it is better to use the same model-free retracker (e.g. OCOG/Ice/Ice1) for both large and small lakes to avoid possible inter-mission retracker-induced biases.

When a lake was visited by more than one satellite missions on the same day, the best water level estimate among the overlapping missions should be selected to form a long-term series of records, in terms of the performance ($r$ and $RMSE$) of the missions.. For the period before 2002, the order of selection priority should be ERS-2, ERS-1 and Topex/Poseidon. For





the period of 2002 – 2013, the order of selection priority should be ENVISat, Jason-2, Jason-1, ERS-2 and GFO. For the

period 2013-2020, the order of selection priority should be Sentinel-3, SARAL, Jason-3 and Jason-2.

**7 Conclusions**

Thirteen satellite radar altimetry missions have been launched to measure the Earth's surface elevation since 1985. The

satellite radar altimetry data collected by these missions have been widely utilized for retrieving lake water levels. Although

some previous studies assessed some missions in retrieving lake water level, our knowledge and understanding are still

limited as to the comparative advantages of different retrackers across different radar altimetry missions and the effective

strategy of tying all missions together to reconstruct a long-term time series to support the investigation of lake water level

dynamics. In this study, we made a comprehensive evaluation on the performances of the different retrackers of eleven

missions using a consistent data processing procedure and algorithms over the same set of twelve case study lakes, where the

*gauge* measurements are available. These twelve lakes are representative for different areal size, local climate and

surrounding environment.

Among the eleven missions, the most recent mission Sentinel-3 gave the most accurate estimates, largely due to the adoption

of new SAR altimetry technology. All three retrackers (particularly the OCOG retracker) of Sentinel-3 yielded very accurate

lake level estimates for both large and small lakes. SARAL's performance is the second best in retrieving lake water levels,

owing to the advantages of the Ka band. Its Ice1 retracker works for both large and small lakes too. ENVISat Ice1 retracker

is slightly better than the Ice retracker of Jason-1 and Jason-2 in terms of $r$ and *RMSE*. However, Jason-1 and -2 can provide

more consistent, frequent and continuous lake water level estimates due to their low data loss rates and short repeat cycle.

Although ERS-1 and ERS-2 (e.g. the Ice1 retracker) had clearly better performance over small lakes than T/P between 1991

and 2005, Topex/Poseiden is still recommended for retrieving water levels for large lakes since it had much more frequent

estimates than ERS-1 and ERS-2. Both GeoSat and GFO exhibited extremely high data loss rates. GFO can be replaced by

several other contemporary missions, such as T/P, ERS-2, Jason-1 and ENVISat. However, GeoSat was the earliest sole

mission in the 1980s, therefore still valuable for extending the time series of lake water level as early as possible.

In order to reconstruct long-term time series of lake water level, a reference mission need to be selected to tie all other

missions together. The best strategy for constructing long-term lake water level records should be a two-step bias correction

and normalization procedure. In the first step, use Jason-2 as the initial reference to estimate the systematic biases with

Topex/Poseidon, Jason-1 and Jason-3 and then normalize them to form a consistent Topex/Poseidon-Jason series. Then, use

Topex/Poseidon-Jason series as the reference to estimate and remove systematic biases with other radar altimetry missions to

construct consistent long-term lake water level series for ungauged lakes. We found that the model-free retrackers

(Ice1/OCOG/Ice) evidently perform better than the model-based retrackers in terms of the *RMSE*, the Pearson's correlation

coefficient $r$ and the data loss rate. For the missions with more than one retrackers, the model-free retracker is recommended

in the construction of the long-term time series of lake water level, particularly, for small lakes. For different time periods,



multiple missions may have overpassed the same lake on the same day. We have worked out the priority order to select the measurements among overlapping missions in three time periods to construct the best possible lake water level time series.

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



**Table 1.** Geographical characteristics of case study lakes and gauge stations

| | Lakes | | | | | | Gauge Stations | |
|---|---|---|---|---|---|---|---|---|
| Index | Name | Country | Lat(°) | Lon(°) | Area (km²) | Winter ice cover | Name | Datum |
| 1 | Inarijarvi | | 69.02 | 27.89 | 1184 | Fully | Nellim | N2000 |
| 2 | Lokka | Finland | 67.96 | 27.63 | 487 | Fully | Lokka | N2000 |
| 3 | Oulujarvi | | 64.35 | 27.21 | 889 | Fully | Vuottolahti | N2000 |
| 4 | Vanern | Sweden | 58.91 | 13.30 | 5550 | None or Partly | Vanern | RH00 |
| 5 | Great Slave | | 61.80 | -113.82 | 27816 | Fully | YellowKnife | CGVD28 |
| 6 | Athabasca | Canada | 59.18 | -109.34 | 7781 | Fully | CrackingStone | CGVD28 |
| 7 | Cedar | | 53.34 | -100.16 | 2817 | Fully | Oleson Point | CGVD28 |
| 8 | Superior | | 47.54 | -87.76 | 81935 | Fully or Partly | Ontonagon | IGLD85 |
| 9 | Huron | US & | 44.96 | -82.26 | 59756 | Partly | Lakeport | IGLD85 |
| 10 | Ontario | Canada | 43.67 | -77.76 | 19328 | Partly | Rochester | IGLD85 |
| 11 | Erie | | 42.16 | -81.24 | 25691 | Fully or Partly | Cleveland | IGLD85 |
| 12 | Michigan | US | 44.01 | -86.76 | 57399 | Partly | Calumet Harbor | IGLD85 |

\* The areas and geographic coordinates are derived from GLWD





**Table 2.** The eleven satellite radar altimetry missions and their data products

| Mission | Operational Time | | Organization | Geographic Coverage | Repeat Cycle | Data Product | | | MWR Bands (GHz) |
| | Begin | End | | | | Source | Rate | Retrackers | |
| --- | --- | --- | --- | --- | --- | --- | --- | --- | --- |
| GeoSat | 03/12/1985 | 01/01/1990 | U.S. Navy | 72°N–72°S | 17 days | RADS | 1 Hz | Ocean | — |
| ERS-1 | 07/17/1991 | 03/10/2000 | ESA | 81.5°N–81.5°S | 35 days | ESA | 20 Hz | Ice1, Ice2, Ocean, Sea-Ice | 23.8, 36.5 |
| T/P | 08/10/1992 | 10/09/2005 | NASA, CNES | 66°N–66°S | 10 days | RADS | 1 Hz | Ocean | 18, 21, 37 |
| ERS-2 | 04/21/1995 | 09/05/2011 | ESA | 81.5°N–81.5°S | 35 days | CTOH | 20 Hz | Ice1, Ice2 | 23.8, 36.5 |
| GFO | 02/10/1998 | 10/22/2008 | U.S. Navy | 72°N–72°S | 17 days | NOAA | 10 Hz | Ocean | 22, 37 |
| Jason-1 | 12/07/2001 | 07/01/2013 | NASA, CNES | 66°N–66°S | 10 days | AVISO+ | 20 Hz | MLE4, Ice | 18.7, 23.8, 34 |
| ENVISat | 02/28/2002 | 04/08/2012 | ESA | 81.5°N–81.5°S | 35 days | ESA | 18 Hz | Ice1, Ice2, Ocean, Sea-Ice | 23.8, 36.5 |
| Jason-2 | 06/20/2008 | 10/01/2019 | NASA, CNES, NOAA, EUMETSAT | 66°N–66°S | 10 days | AVISO+ | 20 Hz | MLE4, Ice | 18.7, 23.8, 34 |
| SARAL | 02/25/2013 | — | CNES, ISRO | 81.5°N–81.5°S | 35 days | AVISO+ | 40 Hz | Ice1, Ice2, Ocean, Sea-Ice | 23.8, 37 |
| Jason-3 | 01/17/2016 | — | NASA, CNES, NOAA, EUMETSAT | 66°N–66°S | 10 days | AVISO+ | 20 Hz | MLE4, Ice | 18.7, 23.8, 34 |
| Sentinel-3 | 02/16/2016 | — | ESA | 81.35°N–81.35°S | 27 days | ESA | 20 Hz | OCOG, Ice-Sheet, Sea-Ice, SAMOSA-3 | 23.8, 36.5 |





**Table 3.** The version of altimetry data product, the models of the three atmospheric corrections.

| Mission | Data Source | Version | Ionospheric Correction ($\Delta R_{iono}$) | Wet Tropospheric Correction ($\Delta R_{wet}$) | Dry Tropospheric Correction ($\Delta R_{dry}$) |
|---|---|---|---|---|---|
| GeoSat | RADS | — | NIC09 | ERA | ERA |
| ERS-1 | ESA | REAPER | NIC09 | ERA | ERA |
| T/P | RADS | GDR-M | NIC09, GIM | ERA | ERA |
| ERS-2 | CTOH | CTOH | NIC09, GIM | ERA | ERA |
| GFO | NOAA | GDR | GIM | NCEP | NCEP |
| Jason-1 | AVISO+ | GDR-E | GIM | ERA | ERA |
| ENVISat | ESA | V3 | GIM | ECMWF | ECMWF |
| Jason-2 | AVISO+ | GDR-D | GIM | ECMWF | ECMWF |
| SARAL | AVISO+ | GDR-T | GIM | ECMWF | ECMWF |
| Jason-3 | AVISO+ | GDR-D | GIM | ECMWF | ECMWF |
| Sentinel-3 | ESA | Baseline-2.45 | GIM | ECMWF | ECMWF |





**Table 4.** The ground tracks selected for each satellite mission over each lake

| Mission | | | | | | | | | | | | | | |
| Name | Phase | Cycles | Inarijarvi | Lokka | Oulujarvi | Vanern | GreatSlave | Athabasca | Cedar | Superior | Huron | Ontario | Erie | Michigan |
| --- | --- | --- | --- | --- | --- | --- | --- | --- | --- | --- | --- | --- | --- | --- |
| GeoSat | Nominal | 67 | 367 | 104 | 397 | 399 | 28 | 228 | 198 | 168 | 293 | 52 | 368 | 24 |
| ERS-1 | Nominal | 32 | 197 | 283 | 541 | 846 | 570 | 409 | 912 | 93 | 80 | 493 | 751 | 93 |
| | Nominal | 333 | — | — | 239 | 144 | 223 | 95 | 178 | 254 | 117 | 15 | 193 | 41 |
| T/P | Intermittent | 111 | — | — | 116 | 220 | 45 | 171 | 195 | 76 | 152 | 50 | 152 | 254 |
| ERS-2 | Nominal | 90 | 197 | 283 | 541 | 846 | 570 | 409 | 912 | 93 | 80 | 493 | 751 | 93 |
| GFO | Nominal | 180 | 367 | 104 | 397 | 399 | 28 | 228 | 198 | 168 | 293 | 52 | 368 | 24 |
| | Nominal | 259 | — | — | 239 | 144 | 223 | 95 | 178 | 254 | 117 | 15 | 193 | 41 |
| Jason-1 | Intermittent | 113 | — | — | 116 | 220 | 45 | 171 | 195 | 76 | 152 | 50 | 152 | 254 |
| ENVISat | Nominal | 89 | 197 | 283 | 541 | 846 | 570 | 409 | 912 | 93 | 80 | 493 | 751 | 93 |
| | Extension | 19 | 169 | 399 | 543 | 648 | 802 | 37 | 312 | 512 | 224 | 581 | 409 | 52 |
| | Nominal | 304 | — | — | 239 | 144 | 223 | 95 | 178 | 254 | 117 | 15 | 193 | 41 |
| Jason-2 | Intermittent | 23 | — | — | 116 | 220 | 45 | 171 | 195 | 76 | 152 | 50 | 152 | 254 |
| SARAL | Nominal | 34 | 197 | 283 | 541 | 846 | 570 | 409 | 912 | 93 | 80 | 493 | 751 | 93 |
| Jason-3 | Nominal | 78 | — | — | 239 | 144 | 223 | 95 | 178 | 254 | 117 | 15 | 193 | 41 |
| Sentinel-3 | Nominal | 18 | 272 | 386 | 72 | 358 | 682 | 54 | 766 | 549 | 422 | 222 | 650 | 736 |

Note that there are no T/P and Jason overpasses for Lake Inarijarvi and Lokka, since their geographic coverage is up to latitude of 66°N.



**Table 5.** The number of lake water level estimates during ice-free (open water) and ice-covered seasons

| Name | Cycles | Retracker | Inarijarvi ice | water | Lokka ice | water | Oulujarvi ice | water | Vanern ice | water | GreatSlave ice | water | Athabasca ice | water | Cedar ice | water | Superior ice | water | Huron ice | water | Ontario ice | water | Erie ice | water | Michigan ice | water |
|---|---|---|---|---|---|---|---|---|---|---|---|---|---|---|---|---|---|---|---|---|---|---|---|---|---|---|
| GeoSat | 67 | Ocean | 16 | 1 | 6 | 1 | 9 | 2 | 5 | 7 | 9 | 10 | 11 | 7 | 0 | 1 | 10 | 28 | 3 | 31 | 0 | 29 | 8 | 41 | 9 | 34 |
| ERS-1 | 32 | Ice1 | 11 | 10 | 14 | 8 | 14 | 11 | 3 | 26 | 14 | 9 | 11 | 13 | 14 | 14 | 5 | 25 | 7 | 18 | 4 | 26 | 4 | 24 | 6 | 26 |
|  |  | Ice2 | 11 | 9 | 14 | 8 | 14 | 11 | 3 | 26 | 14 | 9 | 11 | 13 | 14 | 14 | 5 | 24 | 7 | 19 | 4 | 26 | 4 | 24 | 6 | 26 |
|  |  | Sea-Ice | 11 | 10 | 14 | 8 | 14 | 13 | 3 | 26 | 14 | 9 | 11 | 13 | 14 | 14 | 5 | 24 | 7 | 18 | 4 | 22 | 4 | 23 | 6 | 23 |
|  |  | Ocean | 7 | 7 | 10 | 7 | 10 | 12 | 3 | 26 | 11 | 9 | 7 | 13 | 9 | 12 | 2 | 23 | 4 | 17 | 3 | 22 | 1 | 23 | 4 | 23 |
| T/P | 444 | Ocean | — | — | — | — | 234 | 166 | 15 | 140 | 191 | 122 | 181 | 217 | 183 | 187 | 35 | 300 | 37 | 325 | 43 | 364 | 61 | 313 | 63 | 361 |
| ERS-2 | 90 | Ice1 | 38 | 28 | 30 | 39 | 31 | 21 | 6 | 59 | 35 | 32 | 27 | 34 | 28 | 34 | 12 | 56 | 14 | 64 | 12 | 70 | 12 | 57 | 14 | 53 |
|  |  | Ice2 | 38 | 33 | 30 | 39 | 31 | 21 | 6 | 59 | 35 | 32 | 27 | 34 | 28 | 34 | 12 | 56 | 14 | 64 | 12 | 70 | 12 | 57 | 14 | 53 |
| GFO | 180 | Ocean | 41 | 7 | 92 | 14 | 46 | 7 | 15 | 96 | 70 | 36 | 56 | 42 | 28 | 8 | 9 | 134 | 16 | 107 | 19 | 125 | 21 | 112 | 34 | 125 |
| Jason-1 | 372 | Ice | — | — | — | — | 160 | 149 | 67 | 134 | 206 | 133 | 185 | 193 | 171 | 165 | 42 | 301 | 55 | 293 | 58 | 291 | 31 | 297 | 70 | 279 |
|  |  | MLE4 | — | — | — | — | 88 | 121 | 24 | 90 | 180 | 127 | 176 | 170 | 155 | 118 | 36 | 300 | 48 | 291 | 58 | 291 | 19 | 302 | 70 | 279 |
| ENVISat | 108 | Ice1 | 48 | 45 | 38 | 50 | 38 | 51 | 13 | 85 | 36 | 44 | 45 | 45 | 44 | 45 | 7 | 78 | 11 | 78 | 17 | 84 | 12 | 72 | 15 | 84 |
|  |  | Ice2 | 48 | 43 | 38 | 50 | 38 | 52 | 13 | 85 | 36 | 44 | 45 | 45 | 44 | 45 | 7 | 79 | 11 | 78 | 17 | 84 | 12 | 72 | 14 | 84 |
|  |  | Sea-Ice | 48 | 42 | 38 | 49 | 38 | 41 | 13 | 84 | 36 | 44 | 45 | 45 | 44 | 45 | 7 | 79 | 11 | 78 | 17 | 84 | 12 | 72 | 15 | 84 |
|  |  | Ocean | 48 | 45 | 38 | 50 | 38 | 48 | 13 | 85 | 36 | 44 | 45 | 45 | 44 | 45 | 7 | 75 | 11 | 78 | 17 | 84 | 12 | 72 | 15 | 84 |
| Jason-2 | 327 | Ice | — | — | — | — | 153 | 125 | 54 | 172 | 189 | 128 | 178 | 179 | 172 | 152 | 32 | 282 | 34 | 284 | 58 | 258 | 50 | 272 | 48 | 273 |
|  |  | MLE4 | — | — | — | — | 83 | 102 | 21 | 95 | 161 | 120 | 168 | 162 | 141 | 143 | 28 | 281 | 33 | 283 | 58 | 251 | 37 | 272 | 47 | 273 |
| SARAL | 35 | Ice1 | 16 | 18 | 18 | 16 | 16 | 18 | 1 | 33 | 18 | 16 | 13 | 16 | 16 | 18 | 4 | 29 | 6 | 29 | 6 | 29 | 5 | 29 | 6 | 16 |
|  |  | Ice2 | 16 | 16 | 18 | 16 | 16 | 16 | 1 | 33 | 18 | 16 | 13 | 16 | 16 | 18 | 4 | 30 | 6 | 29 | 6 | 29 | 5 | 29 | 6 | 16 |
|  |  | Sea-Ice | 16 | 17 | 18 | 16 | 16 | 14 | 1 | 33 | 18 | 16 | 13 | 16 | 16 | 18 | 4 | 30 | 6 | 29 | 6 | 29 | 5 | 29 | 6 | 16 |
|  |  | Ocean | 16 | 17 | 18 | 16 | 16 | 15 | 1 | 33 | 18 | 16 | 13 | 16 | 16 | 18 | 4 | 30 | 6 | 29 | 6 | 29 | 5 | 29 | 6 | 16 |
| Jason-3 | 79 | Ice | — | — | — | — | 42 | 27 | 7 | 62 | 43 | 28 | 43 | 35 | 41 | 38 | 8 | 66 | 3 | 72 | 6 | 67 | 8 | 70 | 6 | 69 |
|  |  | MLE4 | — | — | — | — | 23 | 28 | 3 | 36 | 38 | 25 | 37 | 35 | 35 | 35 | 7 | 66 | 3 | 72 | 6 | 66 | 6 | 70 | 6 | 69 |
| Sentinel-3 | 18 | Ice-Sheet | 7 | 10 | 7 | 11 | 7 | 9 | 0 | 18 | 7 | 11 | 7 | 11 | 6 | 10 | 0 | 18 | 0 | 18 | 0 | 18 | 0 | 18 | 0 | 18 |
|  |  | OCOG | 7 | 10 | 7 | 11 | 7 | 9 | 0 | 18 | 7 | 11 | 7 | 11 | 6 | 10 | 0 | 18 | 0 | 18 | 0 | 18 | 0 | 18 | 0 | 18 |
|  |  | SAMOSA3 | 7 | 10 | 7 | 11 | 7 | 9 | 0 | 18 | 7 | 11 | 7 | 11 | 6 | 10 | 0 | 18 | 0 | 18 | 0 | 18 | 0 | 18 | 0 | 18 |






**Table 6.** The data loss rate for lake water level estimates over each lake for each retracker

| Mission | | | Rate of Missing Estimates (%) | | | | | | | | | | | | Mean |
|---|---|---|---|---|---|---|---|---|---|---|---|---|---|---|---|
| Name | Cycles | Retracker | Inarijarvi | Lokka | Oulujarvi | Vanern | GreatSlave | Athabasca | Cedar | Superior | Huron | Ontario | Erie | Michigan | |
| GeoSat | 57 | Ocean | 74.63 | 89.55 | 83.58 | 82.09 | 71.64 | 73.13 | 98.51 | 43.28 | 49.25 | 56.72 | 26.87 | 35.82 | 65.42 |
| ERS-1 | 32 | Ice1 | 34.38 | 31.25 | 21.88 | 9.38 | 28.13 | 25.00 | 12.50 | 6.25 | 21.88 | 6.25 | 12.50 | 0.00 | 17.45 |
| | | Ice2 | 37.50 | 31.25 | 21.88 | 9.38 | 28.13 | 25.00 | 12.50 | 9.38 | 18.75 | 6.25 | 12.50 | 0.00 | 17.71 |
| | | Sea-Ice | 34.38 | 31.25 | 15.63 | 9.38 | 28.13 | 25.00 | 12.50 | 9.38 | 21.88 | 18.75 | 15.63 | 9.38 | 19.27 |
| | | Ocean | 56.25 | 46.88 | 31.25 | 9.38 | 37.50 | 37.50 | 34.38 | 21.88 | 34.38 | 21.88 | 25.00 | 15.63 | 30.99 |
| T/P | 444 | Ocean | — | — | 9.91 | 65.09 | 29.50 | 10.36 | 16.67 | 24.55 | 18.47 | 8.33 | 15.77 | 4.50 | 20.32 |
| ERS-2 | 90 | Ice1 | 26.67 | 23.33 | 42.22 | 27.78 | 25.56 | 32.22 | 31.11 | 24.44 | 13.33 | 8.89 | 23.33 | 25.56 | 25.37 |
| | | Ice2 | 21.11 | 23.33 | 42.22 | 27.78 | 25.56 | 32.22 | 31.11 | 24.44 | 13.33 | 8.89 | 23.33 | 25.56 | 24.91 |
| GFO | 180 | Ocean | 73.33 | 41.11 | 70.56 | 38.33 | 41.11 | 45.56 | 80.00 | 20.56 | 31.67 | 20.00 | 26.11 | 11.67 | 41.67 |
| Jason-1 | 372 | Ice | — | — | 16.94 | 45.97 | 8.87 | 8.06 | 9.68 | 7.80 | 6.45 | 6.18 | 11.83 | 6.18 | 12.80 |
| | | MLE4 | — | — | 43.82 | 69.35 | 17.47 | 12.90 | 26.61 | 9.68 | 8.87 | 6.18 | 13.71 | 6.18 | 21.48 |
| ENVISat | 108 | Ice1 | 13.89 | 18.52 | 17.59 | 9.26 | 25.93 | 16.67 | 17.59 | 21.30 | 17.59 | 6.48 | 22.22 | 8.33 | 16.28 |
| | | Ice2 | 15.74 | 18.52 | 16.67 | 9.26 | 25.93 | 16.67 | 17.59 | 20.37 | 17.59 | 6.48 | 22.22 | 9.26 | 16.36 |
| | | Sea-Ice | 16.67 | 19.44 | 26.85 | 10.19 | 25.93 | 16.67 | 17.59 | 20.37 | 17.59 | 6.48 | 22.22 | 8.33 | 17.36 |
| | | Ocean | 13.89 | 18.52 | 20.37 | 9.26 | 25.93 | 16.67 | 17.59 | 24.07 | 17.59 | 6.48 | 22.22 | 8.33 | 16.74 |
| Jason-2 | 327 | Ice | — | — | 14.98 | 30.89 | 3.06 | 2.75 | 0.92 | 3.98 | 2.75 | 3.36 | 1.53 | 1.83 | 6.61 |
| | | MLE4 | — | — | 43.43 | 64.53 | 14.07 | 7.34 | 13.15 | 5.50 | 3.36 | 5.50 | 5.50 | 2.14 | 16.45 |
| SARAL | 35 | Ice1 | 2.86 | 2.86 | 2.86 | 2.86 | 2.86 | 17.14 | 2.86 | 5.71 | 0.00 | 0.00 | 2.86 | 37.14 | 6.67 |
| | | Ice2 | 8.57 | 2.86 | 8.57 | 2.86 | 2.86 | 17.14 | 2.86 | 2.86 | 0.00 | 0.00 | 2.86 | 37.14 | 7.38 |
| | | Sea-Ice | 5.71 | 2.86 | 14.29 | 2.86 | 2.86 | 17.14 | 2.86 | 2.86 | 0.00 | 0.00 | 2.86 | 37.14 | 7.62 |
| | | Ocean | 5.71 | 2.86 | 11.43 | 2.86 | 2.86 | 17.14 | 2.86 | 2.86 | 0.00 | 0.00 | 2.86 | 37.14 | 7.38 |
| Jason-3 | 79 | Ice | — | — | 12.66 | 12.66 | 10.13 | 1.27 | 0.00 | 6.33 | 5.06 | 7.59 | 1.27 | 5.06 | 6.20 |
| | | MLE4 | — | — | 35.44 | 50.63 | 20.25 | 8.86 | 11.39 | 7.59 | 5.06 | 8.86 | 3.80 | 5.06 | 15.70 |
| Sentinel-3 | 18 | Ice-Sheet | 5.56 | 0.00 | 11.11 | 0.00 | 0.00 | 0.00 | 11.11 | 0.00 | 0.00 | 0.00 | 0.00 | 0.00 | 2.32 |
| | | OCOG | 5.56 | 0.00 | 11.11 | 0.00 | 0.00 | 0.00 | 11.11 | 0.00 | 0.00 | 0.00 | 0.00 | 0.00 | 2.32 |
| | | SAMOSA3 | 5.56 | 0.00 | 11.11 | 0.00 | 0.00 | 0.00 | 11.11 | 0.00 | 0.00 | 0.00 | 0.00 | 0.00 | 2.32 |





**Table 7.** The *Bias* between altimetry-derived estimates and gauge measurements


| Mission | | *Bias* (cm) | | | | | | | | | | | | Mean* | STD* |
|---|---|---|---|---|---|---|---|---|---|---|---|---|---|---|---|
| Name | Retracker | Inarijarvi | Lokka | Oulujarvi | Vanern | GreatSlave | Athabasca | Cedar | Superior | Huron | Ontario | Erie | Michigan | | |
| GeoSat | Ocean | — | — | — | 34.77 | 45.28 | -1.41 | — | 25.10 | 42.36 | 45.59 | 87.61 | 54.20 | 50.97 | 23.05 |
| ERS-1 | Ice1 | 39.23 | 26.05 | 17.43 | 89.40 | 85.69 | 47.43 | 76.69 | 64.18 | 93.36 | 80.85 | 88.40 | 82.14 | 81.79 | 11.05 |
| | Ice2 | 41.60 | 38.26 | 1.18 | 73.43 | 76.48 | 44.64 | 67.18 | 52.13 | 81.08 | 70.66 | 75.67 | 73.48 | 70.60 | 11.01 |
| | Sea-Ice | 38.39 | 17.45 | -20.52 | 62.75 | 76.74 | 43.62 | 64.79 | 53.44 | 77.73 | 69.18 | 73.46 | 71.32 | 69.03 | 9.27 |
| | Ocean | -41.99 | 67.73 | -14.59 | 66.18 | 87.11 | 34.50 | 46.86 | 47.12 | 77.07 | 65.54 | 69.51 | 65.57 | 64.96 | 11.03 |
| T/P | Ocean | — | — | -116.44 | -27.77 | -5.46 | -19.43 | -28.56 | -16.63 | -1.35 | -0.86 | 10.25 | 14.68 | 1.22 | 12.17 |
| ERS-2 | Ice1 | 87.95 | 59.19 | 19.68 | 85.99 | 96.83 | 71.89 | 96.21 | 78.54 | 95.75 | 96.23 | 100.36 | 96.80 | 93.54 | 8.58 |
| | Ice2 | 77.60 | 35.71 | -20.72 | 49.60 | 61.36 | 35.22 | 64.79 | 39.56 | 59.15 | 56.23 | 59.19 | 55.82 | 53.99 | 8.22 |
| GFO | Ocean | -30.49 | -8.05 | -106.08 | 36.98 | 81.16 | 68.83 | 49.58 | 68.81 | 79.90 | 83.97 | 90.97 | 84.35 | 81.60 | 8.18 |
| Jason-1 | Ice | — | — | -47.86 | 18.58 | 23.21 | 11.32 | 20.00 | 10.51 | 26.95 | 26.76 | 35.91 | 39.76 | 27.98 | 11.28 |
| | MLE4 | — | — | -154.03 | 4.36 | 2.44 | -7.85 | -11.17 | -13.08 | 2.45 | 3.68 | 11.88 | 15.44 | 4.07 | 11.04 |
| ENVISat | Ice1 | 25.24 | 18.92 | 11.10 | 75.56 | 73.60 | 49.62 | 59.12 | 59.45 | 74.86 | 77.02 | 79.09 | 74.46 | 72.98 | 7.79 |
| | Ice2 | -3.01 | -0.94 | -17.77 | 51.27 | 48.68 | 25.60 | 39.09 | 34.06 | 47.95 | 49.40 | 52.53 | 47.54 | 46.30 | 7.12 |
| | Sea-Ice | -13.30 | -19.44 | -47.46 | 44.66 | 43.22 | 19.14 | 31.80 | 28.44 | 42.29 | 43.00 | 47.20 | 41.25 | 40.44 | 7.08 |
| | Ocean | 23.45 | 26.75 | -13.65 | 49.78 | 48.22 | 25.43 | 44.58 | 32.31 | 48.48 | 47.87 | 52.17 | 46.52 | 45.47 | 7.65 |
| Jason-2 | Ice | — | — | -28.43 | 36.37 | 44.96 | 36.08 | 18.60 | 19.81 | 48.37 | 49.24 | 57.12 | 60.64 | 47.04 | 16.08 |
| | MLE4 | — | — | -165.82 | -31.22 | 5.68 | -4.12 | -13.54 | -14.50 | 1.29 | 4.33 | 11.34 | 14.55 | 3.40 | 11.33 |
| SARAL | Ice1 | 38.85 | 28.92 | 33.60 | 84.11 | 79.71 | 79.46 | 58.24 | 81.61 | 77.72 | 97.40 | 101.24 | 94.69 | 90.53 | 10.28 |
| | Ice2 | 25.77 | 26.31 | 22.50 | 70.74 | 67.02 | 65.49 | 49.05 | 67.39 | 62.30 | 84.32 | 85.00 | 79.91 | 75.78 | 10.33 |
| | Sea-Ice | 11.05 | 3.08 | -7.18 | 64.07 | 60.73 | 58.56 | 40.20 | 60.23 | 54.30 | 77.74 | 77.48 | 72.82 | 68.51 | 10.66 |
| | Ocean | 21.30 | -5.71 | -13.91 | 75.36 | 71.05 | 50.69 | 49.22 | 51.57 | 66.04 | 69.78 | 69.24 | 64.81 | 64.29 | 7.41 |
| Jason-3 | Ice | — | — | -46.73 | 11.14 | 23.82 | 20.14 | 0.81 | -0.82 | 26.32 | 29.03 | 36.35 | 39.99 | 26.17 | 16.06 |
| | MLE4 | — | — | -189.09 | 0.98 | 6.78 | -2.91 | -12.36 | -16.63 | -1.70 | 2.28 | 7.19 | 11.26 | 0.48 | 10.75 |
| Sentinel-3 | Ice-Sheet | -36.19 | -19.60 | -33.72 | 21.27 | 19.64 | -0.40 | 13.57 | 1.52 | 11.96 | 3.16 | 17.90 | 9.50 | 8.81 | 6.67 |
| | OCOG | -25.99 | -34.81 | -16.17 | 45.17 | 42.47 | 23.08 | -2.90 | 26.72 | 37.95 | 28.60 | 42.93 | 35.22 | 34.28 | 6.68 |
| | SAMOSA3 | -37.01 | -19.33 | -42.15 | 14.18 | 12.95 | -7.00 | 4.08 | -6.35 | 4.86 | -3.96 | 10.94 | 2.26 | 1.55 | 6.94 |

* The mean *Bias* and the standard deviation (STD) were computed for each retracker using only the *Biases* on the Great Lakes.





**Table 8.** The Pearson's correlation coefficient *r* between altimetry-derived lake level estimates and *gauge* measurements

| Mission Name | Retracker | Inarijarvi | Lokka | Oulujarvi | Vanern | GreatSlave | Athabasca | Cedar | Superior | Huron | Ontario | Erie | Michigan | Mean |
|---|---|---|---|---|---|---|---|---|---|---|---|---|---|---|
| GeoSat | Ocean | — | — | — | 0.45 | 0.13 | 0.84 | — | 0.78 | 0.94 | 0.93 | 0.88 | 0.93 | 0.74 |
| ERS-1 | Ice1 | 0.56 | 0.79 | 0.80 | 0.97 | 0.98 | 0.94 | 0.94 | 0.76 | 0.94 | 0.97 | 0.91 | 0.76 | 0.86 |
| | Ice2 | 0.29 | 0.62 | 0.65 | 0.84 | 0.73 | 0.93 | 0.66 | 0.69 | 0.61 | 0.94 | 0.88 | 0.80 | 0.72 |
| | Sea-Ice | 0.35 | 0.66 | 0.49 | 0.63 | 0.41 | 0.75 | 0.70 | 0.10 | 0.35 | 0.71 | 0.43 | 0.47 | 0.50 |
| | Ocean | 0.70 | 0.68 | 0.07 | 0.90 | 0.73 | 0.94 | 0.38 | 0.83 | 0.72 | 0.97 | 0.92 | 0.87 | 0.73 |
| T/P | Ocean | — | — | 0.21 | 0.58 | 0.84 | 0.97 | 0.93 | 0.97 | 0.99 | 0.98 | 0.98 | 0.99 | 0.84 |
| ERS-2 | Ice1 | 0.62 | 0.89 | 0.57 | 0.95 | 0.93 | 0.99 | 0.96 | 0.79 | 0.96 | 0.91 | 0.96 | 0.96 | 0.87 |
| | Ice2 | 0.24 | 0.76 | 0.41 | 0.94 | 0.87 | 0.99 | 0.90 | 0.75 | 0.95 | 0.86 | 0.93 | 0.95 | 0.80 |
| GFO | Ocean | 0.95 | 0.47 | 0.14 | 0.95 | 0.91 | 0.97 | 0.80 | 0.92 | 0.93 | 0.98 | 0.95 | 0.92 | 0.82 |
| Jason-1 | Ice | — | — | 0.77 | 0.89 | 0.94 | 0.98 | 0.91 | 0.96 | 0.94 | 0.99 | 0.95 | 0.93 | 0.93 |
| | MLE4 | — | — | 0.22 | 0.89 | 0.94 | 0.99 | 0.93 | 0.96 | 0.95 | 0.99 | 0.96 | 0.93 | 0.88 |
| ENVISat | Ice1 | 0.82 | 0.99 | 0.94 | 0.99 | 0.96 | 0.99 | 0.98 | 0.97 | 0.94 | 0.97 | 0.97 | 0.88 | 0.95 |
| | Ice2 | 0.72 | 0.98 | 0.78 | 0.99 | 0.96 | 0.99 | 0.97 | 0.97 | 0.96 | 0.98 | 0.98 | 0.94 | 0.94 |
| | Sea-Ice | 0.46 | 0.97 | 0.69 | 0.98 | 0.92 | 0.98 | 0.95 | 0.96 | 0.95 | 0.97 | 0.98 | 0.93 | 0.90 |
| | Ocean | 0.46 | 0.91 | 0.68 | 0.97 | 0.86 | 0.98 | 0.85 | 0.98 | 0.90 | 0.97 | 0.98 | 0.93 | 0.87 |
| Jason-2 | Ice | — | — | 0.75 | 0.85 | 0.92 | 0.99 | 0.94 | 0.98 | 0.97 | 0.98 | 0.96 | 0.97 | 0.93 |
| | MLE4 | — | — | 0.29 | 0.18 | 0.95 | 0.99 | 0.88 | 0.98 | 0.98 | 0.99 | 0.98 | 0.98 | 0.82 |
| SARAL | Ice1 | 0.87 | 0.93 | 0.93 | 0.98 | 0.97 | 0.99 | 0.89 | 0.97 | 0.96 | 0.99 | 0.94 | 0.98 | 0.95 |
| | Ice2 | 0.85 | 0.92 | 0.87 | 0.99 | 0.99 | 0.99 | 0.72 | 0.98 | 0.98 | 0.99 | 0.99 | 0.99 | 0.94 |
| | Sea-Ice | 0.85 | 0.89 | 0.67 | 0.99 | 0.99 | 0.99 | 0.81 | 0.99 | 0.99 | 0.99 | 0.99 | 0.99 | 0.93 |
| | Ocean | 0.75 | 0.90 | 0.56 | 0.99 | 0.99 | 0.99 | 0.79 | 0.98 | 0.97 | 0.99 | 0.98 | 0.99 | 0.91 |
| Jason-3 | Ice | — | — | 0.52 | 0.66 | 0.93 | 0.99 | 0.95 | 0.97 | 0.95 | 0.99 | 0.97 | 0.95 | 0.89 |
| | MLE4 | — | — | 0.53 | 0.24 | 0.94 | 0.99 | 0.93 | 0.95 | 0.95 | 0.99 | 0.96 | 0.93 | 0.84 |
| Sentinel-3 | Ice-Sheet | 0.98 | 0.86 | 0.74 | 0.98 | 0.97 | 0.99 | 0.81 | 0.95 | 0.98 | 0.99 | 0.99 | 0.96 | 0.93 |
| | OCOG | 0.99 | 0.90 | 0.98 | 0.98 | 0.97 | 0.99 | 0.86 | 0.97 | 0.98 | 0.99 | 0.98 | 0.95 | 0.96 |
| | SAMOSA3 | 0.99 | 0.89 | 0.90 | 0.99 | 0.97 | 0.99 | 0.78 | 0.91 | 0.98 | 0.99 | 0.99 | 0.96 | 0.94 |





**Table 9.** The *RMSE* between altimetry-derived lake water level estimates and gauge measurements

| Mission | | \multicolumn RMSE (cm) | | | | | | | | | | | | Mean | STD |
|---|---|---|---|---|---|---|---|---|---|---|---|---|---|---|---|
| Name | Retracker | Inarijarvi | Lokka | Oulujarvi | Vanern | GreatSlave | Athabasca | Cedar | Superior | Huron | Ontario | Erie | Michigan | Mean | STD |
| GeoSat | Ocean | — | — | — | 23.74 | 45.28 | 10.00 | — | 8.80 | 9.19 | 9.03 | 13.08 | 10.09 | 16.15 | 11.95 |
| ERS-1 | Ice1 | 53.16 | 33.01 | 17.91 | 4.75 | 5.49 | 9.16 | 9.72 | 10.54 | 5.30 | 8.08 | 9.39 | 10.65 | 14.76 | 3.05 |
| | Ice2 | 78.90 | 49.05 | 19.17 | 14.77 | 18.02 | 10.28 | 38.01 | 8.93 | 13.55 | 9.88 | 9.63 | 9.71 | 23.33 | 9.12 |
| | Sea-Ice | 73.47 | 52.14 | 35.42 | 31.75 | 33.87 | 22.95 | 37.75 | 24.76 | 31.32 | 26.80 | 26.65 | 25.20 | 35.17 | 5.02 |
| | Ocean | 95.90 | 21.78 | 94.89 | 9.97 | 42.36 | 11.09 | 89.19 | 6.76 | 11.10 | 8.02 | 7.57 | 7.89 | 33.88 | 26.42 |
| T/P | Ocean | — | — | 66.33 | 38.78 | 11.44 | 12.49 | 17.41 | 4.53 | 6.35 | 5.48 | 5.61 | 7.08 | 17.55 | 10.37 |
| ERS-2 | Ice1 | 24.58 | 30.84 | 28.21 | 14.09 | 9.72 | 8.70 | 13.86 | 14.55 | 14.29 | 12.79 | 11.90 | 13.51 | 16.42 | 2.32 |
| | Ice2 | 50.67 | 59.77 | 46.69 | 15.08 | 12.32 | 9.04 | 23.67 | 15.25 | 14.23 | 15.53 | 13.62 | 15.50 | 24.28 | 4.71 |
| GFO | Ocean | 51.78 | 101.20 | 132.81 | 8.64 | 7.17 | 10.73 | 33.65 | 6.03 | 6.21 | 5.09 | 5.54 | 6.61 | 31.29 | 10.88 |
| Jason-1 | Ice | — | — | 17.42 | 8.53 | 5.38 | 6.48 | 47.91 | 4.29 | 6.00 | 4.41 | 5.72 | 6.68 | 11.28 | 13.25 |
| | MLE4 | — | — | 124.98 | 8.67 | 5.55 | 4.33 | 17.57 | 4.65 | 5.37 | 3.56 | 5.26 | 6.88 | 18.68 | 5.50 |
| ENVISat | Ice1 | 16.26 | 14.69 | 12.66 | 3.16 | 4.41 | 5.15 | 7.55 | 3.95 | 6.08 | 5.92 | 5.09 | 8.01 | 7.74 | 1.67 |
| | Ice2 | 20.22 | 16.30 | 27.01 | 3.32 | 4.97 | 4.90 | 9.60 | 4.40 | 4.55 | 4.33 | 3.87 | 6.00 | 9.12 | 2.15 |
| | Sea-Ice | 32.45 | 24.77 | 30.35 | 4.15 | 6.83 | 6.36 | 11.64 | 5.25 | 5.11 | 5.81 | 4.33 | 6.52 | 11.96 | 2.78 |
| | Ocean | 35.25 | 40.17 | 29.14 | 4.83 | 9.48 | 7.73 | 21.25 | 4.06 | 7.51 | 5.85 | 4.16 | 6.49 | 14.66 | 5.44 |
| Jason-2 | Ice | — | — | 17.61 | 11.97 | 7.06 | 6.41 | 7.05 | 4.37 | 7.31 | 5.28 | 6.79 | 7.92 | 8.18 | 2.02 |
| | MLE4 | — | — | 99.91 | 44.94 | 5.14 | 6.88 | 10.87 | 5.17 | 5.28 | 3.59 | 4.64 | 5.81 | 19.22 | 12.74 |
| SARAL | Ice1 | 15.41 | 23.69 | 6.77 | 3.28 | 4.91 | 4.09 | 6.00 | 4.25 | 10.23 | 3.44 | 7.62 | 4.96 | 7.89 | 2.26 |
| | Ice2 | 13.69 | 26.13 | 10.05 | 2.05 | 3.26 | 2.91 | 9.93 | 3.60 | 6.78 | 1.92 | 3.23 | 4.02 | 7.30 | 2.62 |
| | Sea-Ice | 12.56 | 35.21 | 23.96 | 2.50 | 2.76 | 2.98 | 7.94 | 3.02 | 5.39 | 2.30 | 3.48 | 4.14 | 8.85 | 2.33 |
| | Ocean | 26.11 | 29.31 | 28.23 | 2.80 | 3.39 | 4.00 | 9.39 | 3.28 | 7.88 | 2.63 | 4.02 | 4.47 | 10.46 | 2.89 |
| Jason-3 | Ice | — | — | 24.65 | 17.20 | 5.65 | 4.44 | 6.05 | 3.19 | 4.76 | 4.15 | 4.89 | 5.36 | 8.03 | 4.02 |
| | MLE4 | — | — | 110.32 | 24.02 | 3.78 | 4.51 | 4.00 | 4.83 | 5.00 | 3.24 | 5.02 | 6.82 | 17.15 | 6.98 |
| Sentinel-3 | Ice-Sheet | 11.24 | 28.38 | 12.04 | 2.58 | 3.43 | 3.04 | 9.11 | 4.81 | 4.26 | 2.69 | 2.26 | 3.89 | 7.31 | 2.23 |
| | Ice1 | 9.35 | 21.46 | 5.17 | 2.12 | 3.25 | 2.56 | 11.13 | 3.61 | 4.51 | 1.45 | 3.77 | 4.60 | 6.08 | 2.74 |
| | SAMOSA3 | 8.25 | 23.68 | 5.14 | 2.85 | 3.79 | 3.75 | 11.07 | 6.22 | 4.48 | 3.06 | 2.56 | 4.01 | 6.57 | 2.56 |






**Figure 1**. Geographic distribution of the case study lakes. Lakes are labelled with an identification number listed in Table 1. This figure is adapted from Fig 1 in (Shu et al., 2020).






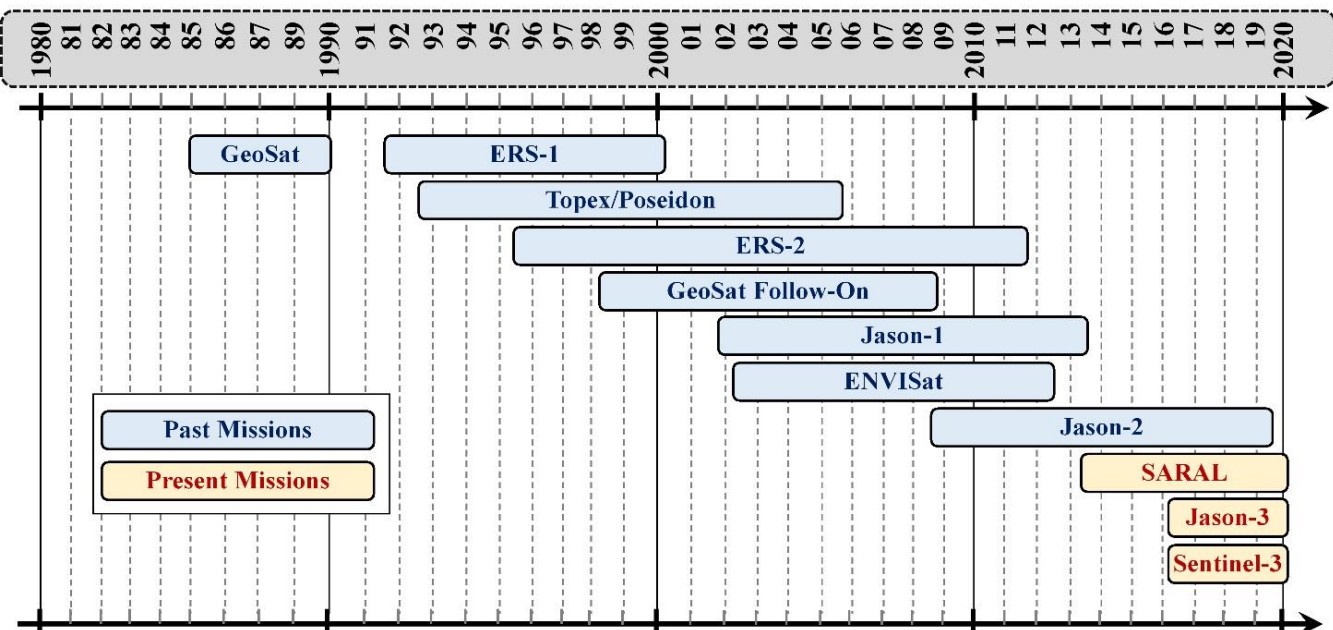

**Figure 2.** Timeline of the eleven satellite radar altimetry missions






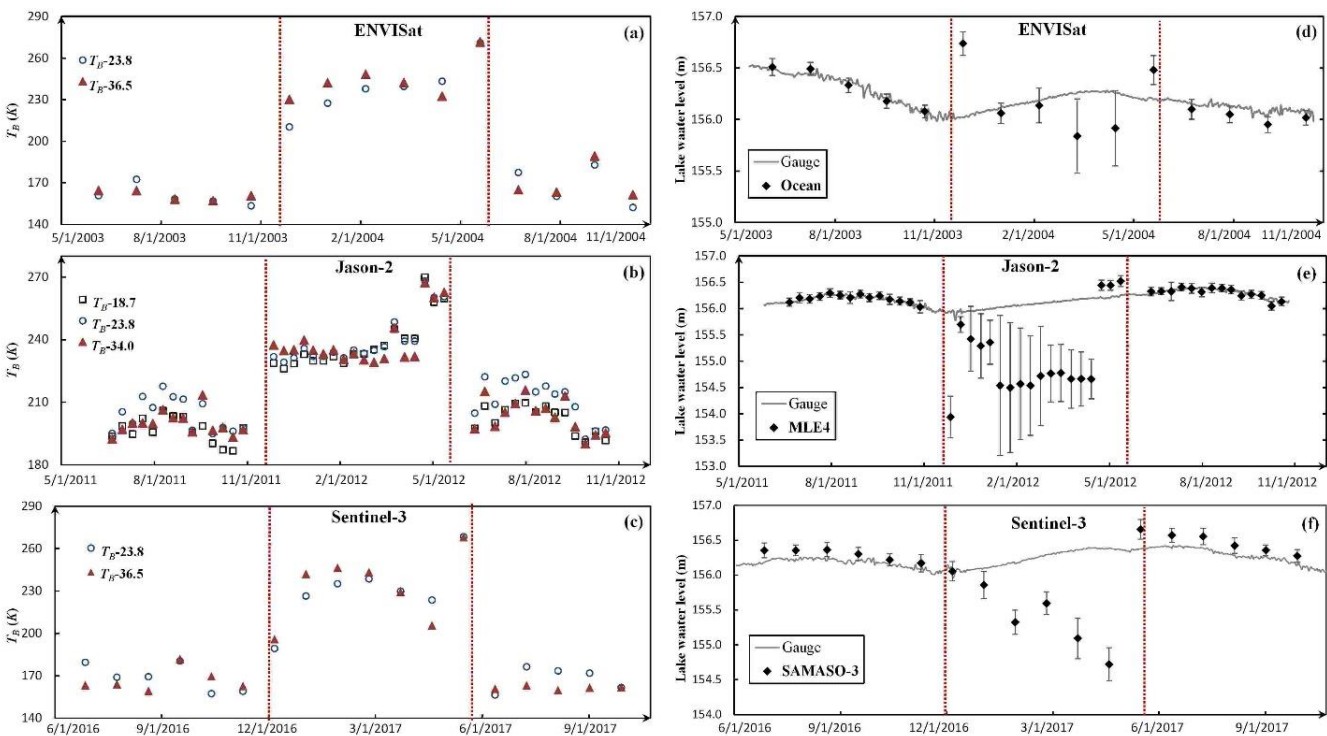

**Figure 3**. Time series of mean brightness temperature ($T_B$) and lake level estimates produced by ENVISat, Jason-2 and Sentinel-3 over Great Slave Lake in the winters. (a) $T_B$ in 2003/2004 winter, (b) $T_B$ in 2011/2012 winter, (c) $T_B$ in 2016/2017 winter, (d) lake level estimate in 2003/2004 winter, (e) lake level estimates in 2011/2012 winter; and (f) lake level estimates in 2016/2017 winter.






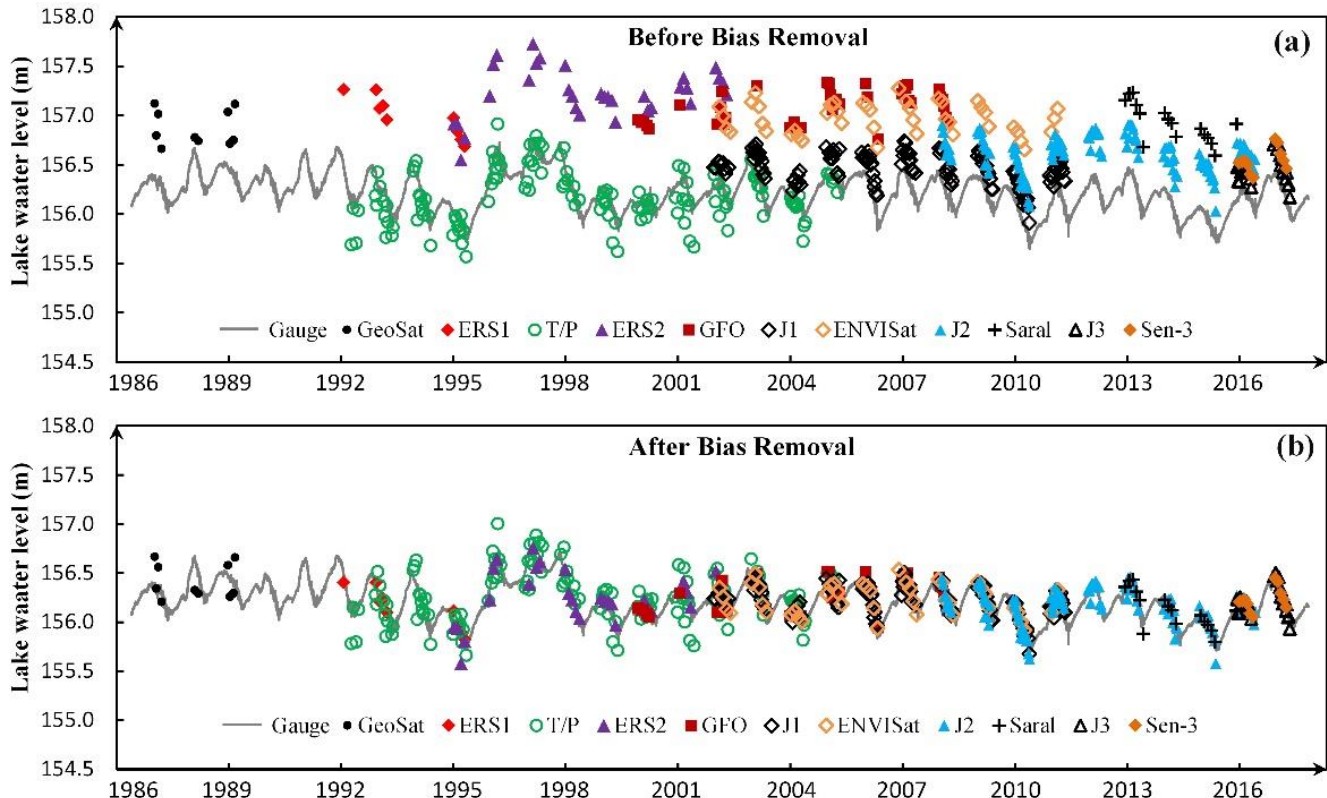

**Figure 4.** The long-term time series of lake water level derived from the eleven satellite radar altimetry missions for Great Slave Lake in Canada; (a) the *Biases* between altimetry-derived estimates and *gauge* measurements were not removed; (b) the *Biases* were removed.





**Figure 5**. Scatterplots of lake water level estimates given by model-free retrackers against *gauge* measurements. The scatterplots of the same mission are arranged in the same row and the scatterplots of the same lake are arranged in the same column.

**Figure 6**. Scatterplots of lake water level estimates given by model-based retrackers against *gauge* measurements. The scatterplots of the same mission are arranged in the row and the scatterplots of the same lake are arrange in the column.