# Peer review of "Evaluation of Historic and Operational Satellite Radar Altimetry Missions for Constructing Consistent Long-term Lake Water Level Records"

_Hydrology and Earth System Sciences, 2020_

## Referee Comment (RC1) · Anonymous Referee #1 · 19 Nov 2020

The paper provides a detailed analysis of past & present altimetry mission data for lake water level retrieval. The paper is well written & organised and results are correctly presented and discussed. Official retrackers are discussed in detail underlining pros & cons. A strategy for constructing for a consistent long-term lake water level is presented. If implemented, it would have added a significant contribution to the paper. Hopefully this is something that authors will present in a future paper. I recommend to accept the paper implementing the minor changes reported below. Some points could be discussed with more detail but this is essentially a very good paper deserving to be published.

**General comments:**

- SAMOSA3 is cited everywhere as the official S3 ocean retracker. This is not correct as it has been updated a long time ago to SAMOSA2 (https://sentinel.esa.int/documents/247904/2802412/Sentinel-3-Mission-Status-Report-06-December-2017.pdf). In table 3, authors indicated Baseline 2.45, this confirms that the SAMOSA 2.5 model (SAMOSA2) has been used as it was introduced in Processing Baseline 2.24 according to the Labroue et al. talk at the 2018 S3VT meeting in Darmstadt. Please correct from "SAMOSA3" to SAMOSA2" everywhere in the manuscript.

- Please correct from ENVIsat to ENVISAT everywhere.

- In this work official retrackers have been considered, however, many non-official efficient retrackers have been developed for the inland water domain (SAMOSA+, DTU MWaPP, please see and cite the following as well:   https://doi.org/10.1016/j.rse.2019.111546) performing better than OCOG. Therefore, for a possible future paper, we suggest the authors to test these alternative retrackers against the OCOG for S3 and also report the results at the Sentinel-3 Validation Team Meeting in order to eventually stimulate the adoption of retrackers alternative to OCOG. The same should be done by citing GPD+ Tropo corrections which many papers indicate as a valid alternative for the inland water domain.

- The discussion on input datasets is very good whereas the complexity of the scenario is not discussed with the same level of detail in relation to the surrounding topography (with respect to lake size), tracking modes (open loop/closed loop) & size of the receiving window. This is an important point considered in the majority of papers investigating the performance of altimetry systems in the inland water domain.

- In the conclusions, the FF-SAR could be cited for future investigations (see https://doi.org/10.1016/j.rse.2019.111589 ) as services  will be providing FF-SAR Sentinel-3 and Cryosat-2 data shortly (Scagliola et al. 2020 in OSTST2020, Moreau et al. in the 2020 Coastal Altimetry Workshop final report available at http://doi.org/10.5270/esa.caw12_2020.final_report.). Sentinel-6 data can also be processed in Fully Focused mode when available.

**Specific comments**

**Abstract section:**

For Sentinel-3, Tables 9 and 8 indicate that the mean results are equivalent for the OCOG and SAMOSA, this should be underlined in the discussion. The bias (Table 7) is way lower for Sentinel-3. Therefore, the statement in the abstract (*"The results show that the model-free retrackers (e.g. OCOG/Ice-1/Ice) outperform the model-based retrackers for all missions, particularly over small lakes."*) shall be revised.

**Introduction:**

- In citing each mission, a reference paper should be added.

- Please indicate that Cryosat-2 is able operating the SARin mode in: *"Most of the radar altimeters operate in a conventional low-resolution mode (LRM), whereas Sentinel-3 and Cryosat-2 operate in Synthetic Aperture Radar (SAR) mode."*

- Please support the following with a reference: *"the River and Lake database (http://www.cse.dmu.ac.uk/EAPRS/products_riverlake.html) built by the ESA and De Montfort University (ESA-DMU),"* as made for the other databases.

- The position of this reference: *"Jarihani et al. (2013) compared five different satellite […]"* shall be revised in the references list as the name is reported before the surname:

> Asadzadeh Jarihani, A., Callow, J. N., Johansen, K., and Gouweleeuw, B.: Evaluation of multiple satellite altimetry data for studying inland water bodies and river floods, Journal of Hydrology, 505, 78-90, 10.1016/j.jhydrol.2013.09.010, 2013.

- Please evaluate revising from *"self-developed retrackers"* to *"non-official retrackers"* when citing Villadsen et al. (2016). Please cite also this paper in the sentence: https://doi.org/10.1016/j.rse.2019.111546.

- The following could be a bit more detailed: *"HY-2A was excluded from this study because of the difficulty in obtaining its data product."*

**Section 2.1**

- The overall discussion on ice cover and presence of small islands is fine. Can something more be said about the complexity of the topography surrounding each of the investigated lake? This should be related to the tracking modes (open loop/closed loop) & size of the receiving window of the specific altimetry system to enhance the discussion. This is a very important point which is not discussed in detail in the paper (e.g. ENVISAT operated with 3 possible bandwidths/receiving window sizes allowing the instrument to correctly operate on various surfaces). This could be related to the data loss rate discussed in Table 6.

**Section 3**

- Table 2 is introduced with the following sentence: *"We used the most up-to-date version of data product of each mission for the evaluation. The geographical coverage, operational time period, repeat cycle, footprint size and retrackers of these radar altimetry missions are summarized in Table 2.'*. Please correct from *"footprint size"* to posting rate:

| Mission | Operational Time Begin | Operational Time End | Organization | Geographic Coverage | Repeat Cycle | Data Product Source | Data Product Rate | Data Product Retrackers | MWR Bands (GHz) |
|---|---|---|---|---|---|---|---|---|---|
| GeoSat | 03/12/1985 | 01/01/1990 | U.S. Navy | 72°N–72°S | 17 days | RADS | 1 Hz | Ocean | — |
| ERS-1 | 07/17/1991 | 03/10/2000 | ESA | 81.5°N–81.5°S | 35 days | ESA | 20 Hz | Ice1, Ice2, Ocean, Sea-Ice | 23.8, 36.5 |
| T/P | 08/10/1992 | 10/09/2005 | NASA, CNES | 66°N–66°S | 10 days | RADS | 1 Hz | Ocean | 18, 21, 37 |
| ERS-2 | 04/21/1995 | 09/05/2011 | ESA | 81.5°N–81.5°S | 35 days | CTOH | 20 Hz | Ice1, Ice2 | 23.8, 36.5 |
| GFO | 02/10/1998 | 10/22/2008 | U.S. Navy | 72°N–72°S | 17 days | NOAA | 10 Hz | Ocean | 22, 37 |
| Jason-1 | 12/07/2001 | 07/01/2013 | NASA, CNES | 66°N–66°S | 10 days | AVISO+ | 20 Hz | MLE4, Ice | 18.7, 23.8, 34 |
| ENVISat | 02/28/2002 | 04/08/2012 | ESA | 81.5°N–81.5°S | 35 days | ESA | 18 Hz | Ice1, Ice2, Ocean, Sea-Ice | 23.8, 36.5 |
| Jason-2 | 06/20/2008 | 10/01/2019 | NASA, CNES, NOAA, EUMETSAT | 66°N–66°S | 10 days | AVISO+ | 20 Hz | MLE4, Ice | 18.7, 23.8, 34 |
| SARAL | 02/25/2013 | — | CNES, ISRO | 81.5°N–81.5°S | 35 days | AVISO+ | 40 Hz | Ice1, Ice2, Ocean, Sea-Ice | 23.8, 37 |
| Jason-3 | 01/17/2016 | — | NASA, CNES, NOAA, EUMETSAT | 66°N–66°S | 10 days | AVISO+ | 20 Hz | MLE4, Ice | 18.7, 23.8, 34 |
| Sentinel-3 | 02/16/2016 | — | ESA | 81.35°N–81.35°S | 27 days | ESA | 20 Hz | OCOG, Ice-Sheet, Sea-Ice, SAMOSA-3 | 23.8, 36.5 |

- *Please improve including 'empirical' & 'physical' in the following sentence: "These retrackers can be divided into two general categories: the* **empirical/***model-free retrackers and the* **physical/***model-based retrackers."*

- *Typo, plural in "retrackers": "and the Sentinel-3 Ice-Sheet retracker is based on a 5-part piecewise analytical function (MSSL/UCL/CLS, 2019)."*

- *Regarding "Jason-3 now operates on the nominal orbit and will continue until the planned launch of Jason-CS/Sentinel-6 in 2020." Please replace "Jason-CS/Sentinel-6" with "Sentinel-6 Michael Freilich"*

**Section 4**

- Typo "e" in : "and the most recent release e of the altimetry"

- Regarding *"Third, the ice-cover condition is examined using the simultaneous TB measurements from the MWR instruments, and those lake water level estimates during the ice-covered period are excluded in the subsequent accuracy evaluations."* Did author consider the possibility of comparing TB measurements results to ice charts?

**Section 4.1**

- Please indicate "orthometric height" in "Geoid converts the reference surface from 330 ellipsoid to geoid**(orthometric height)**".

**Section 6**

- Regarding *"Our evaluation result is contrary to Sulistioadi (2015), who found comparable performances between Sea Ice and OCOG retrackers over a couple of small lakes using ENVISat data." Please do not be generic and clearly name the lakes studied in Sulistioadi* **et. al** *(2015). As previously indicated, one cannot exclude that other factors*

*(e.g. topography) & filtering criteria played a role in justifying the results obtained by Sulistioadi **et. al** (2015). To confirm that the OCOG is better, your analysis should be done over the same lakes and the methodologies adopted compared in discussing the results.*

- A possible strategy to create a multi mission time series is discussed. If implemented, it would have added a significant contribution to the paper.

- *Regarding: "When a lake was visited by more than one satellite missions on the same day, the best water level estimate among the overlapping missions should be selected to form a long-term series of records, in terms of the performance (r and RMSE) of the missions", which criteria would authors suggest to select the "the best water level estimate among the overlapping missions"?*

- *Typo (double full stop)in: "[…] in terms of the performance (r and RMSE) of the missions.."*

- *Please modify, according to table 9, from "6.47" to "6.08" in "The mean RMSE decreases from 35.17 cm of the early ERS-1 mission to 6.47 cm of the current Sentinel-3 mission."*

**On Tables & Figures**

**Table 2**

- Sentinel-3 is indicated with a single launch date. Please consider including 2 entries for both Sentinel-3A & Sentinel-3B.

**Table 4**

- Please explain how (see Sentinel-3 for example) for 18 Cycles you have 272 ground tracks selected for the first lake in the table.

---

## Referee Comment (RC2) · Anonymous Referee #2 · 25 Nov 2020

The paper evaluates water levels based on almost all historic and current altimetry missions and their standard retrackers over 12 lakes of different sizes. Here, especially, the results of the older missions are interesting. The main issue with this paper is the small sample size. 12 lakes are too small to provide any solid recommendations. Having a larger and more representative sample size would make this paper much more valuable. The Paper is well written and organized. The paper can be accepted if the review comment is addressed. Here, especially a discussion of low sample size is needed and the conclusions should be modified accordingly.

General comments: To make solid statements and recommendations about the rekraking performance, 12 lakes are too small a sample size. This should at least be mentioned in the discussions section. However, the results in the paper support similar results in the literature.

The method section is vague and must be extended so it at least summarizes the methods from the mentioned reference studies. Hence, The MAD is estimated but what is the threshold to reject an observation.

A main point in the paper is to construct consistent long-term time series and one of the issues is the intermission/retracking bias. In section 5.2 the gauge is used to estimate the biases. However, as discussed in the Discussion Section a gauge is not always available and therefore the bias should be estimated relative to a reference(s) mission. Why did the authors not test this approach?

Why do the authors select evaluation targets in ice-covered regions when measurements during ice-covered periods are removed anyway?

Why only use 1 track from each mission in the time series if more are available, this would improve the temporal resolution and the statistical foundation. Anyway, some of the missions are in different orbits anyway. For this reason, C2 could also have been included. Several authors have successfully applied C2 for lake level estimation.

Specific Comments

L296: Shu et al, 2020 is not the reference of the standard S3A retrackers.

L306: why only use such a small time period of S3 and Jason-3 in the evaluation?

L331: add a reference to EGM2008

L361: which criterion is used to remove outliers

L364: "through" -> over

L393: The r indicates -> the Pearson correlation r ...

L420: When you calculate the data loss rate is that based on the "valid" measurements or all measurements

L440: This only makes sense to state if the gauge and altimetry has the same vertical reference

L448: is the bias calculated w.r.t the gauge? then add this

L495-503: Put all the numbers in a table

L510: Such conclusions are difficult to state based on just a few lakes

L582: How would you determine which mission provides the best measurement?

---

## Author Comment (AC1) · 19 Jan 2021

Our Response to Anonymous Referee 1

The paper provides a detailed analysis of past present altimetry mission data for lake water level retrieval. The paper is well written organised and results are correctly presented and discussed. Official retrackers are discussed in detail underlining pros cons. A strategy for constructing for a consistent long-term lake water level is presented. If implemented, it would have added a significant contribution to the paper. Hopefully this is something that authors will present in a future paper. I recommend to accept the paper implementing the minor changes reported below. Some points could be discussed

with more detail but this is essentially a very good paper deserving to be published.

Response: We thank the reviewer for the thorough review and very helpful comments/suggestions. The positive comments encourage us to continue working on this subject, particularly the construction of consistent long-term lake water levels at regional or global scale in the future.

General comments: - SAMOSA3 is cited everywhere as the official S3 ocean retracker. This is not correct as it has been updated a long time ago to SAMOSA2 (https://sentinel.esa.int/documents/247904/2802412/Sentinel-3-Mission-Status-Report-06-December-2017.pdf). In table 3, authors indicated Baseline 2.45, this confirms that the SAMOSA 2.5 model (SAMOSA2) has been used as it was introduced in Processing Baseline 2.24 according to the Labroue et al. talk at the 2018 S3VT meeting in Darmstadt. Please correct from "SAMOSA3" to SAMOSA2" everywhere in the manuscript.

Response: We really appreciate the reviewer's valuable information. We have corrected "SAMOSA3" to SAMOSA2" throughout the manuscript, including the text, tables and figures.

- Please correct from ENVIsat to ENVISAT everywhere.

Response: revised as the reviewer suggested.

- In this work official retrackers have been considered, however, many non-official efficient retrackers have been developed for the inland water domain (SAMOSA+, DTU MWaPP, please see and cite the following as well: https://doi.org/10.1016/j.rse.2019.111546) performing better than OCOG. Therefore, for a possible future paper, we suggest the authors to test these alternative retrackers against the OCOG for S3 and also report the results at the Sentinel-3 Validation Team Meeting in order to eventually stimulate the adoption of retrackers alternative to OCOG. The same should be done by citing GPD+ Tropo corrections which many

papers indicate as a valid alternative for the inland water domain.

Response: Thanks for the reviewer's suggestion and information. The suggested papers and sources are cited in the revised manuscript. We will surely include the evaluation of these non-official retrackers in our future study.

- The discussion on input datasets is very good whereas the complexity of the scenario is not discussed with the same level of detail in relation to the surrounding topography (with respect to lake size), tracking modes (open loop/closed loop) size of the receiving window. This is an important point considered in the majority of papers investigating the performance of altimetry systems in the inland water domain.

Response: We appreciate the reviewer's insightful comments. These factors are indeed very important for retrieving water level over small lakes (width less than 1 or 2 km) or over rivers. Particularly, the tracking modes (open/closed loops) and the receiving window could have considerable influences on the accuracy of water level estimates when the surrounding topography is complex. In response to the reviewer's comments, we added that the surrounding topography could have nonnegligible influences on elevation measurements in Section 4.2. The smallest case study lake in our evaluation is Reservoir Lokka in Finland with a surface area of about 500 km2. For each mission, the ground track over the lake is at least 10 kilometers long. To eliminate the possible influence of surrounding topography and land contamination. We have removed the observations within 2 km buffer distance from the shoreline. In addition, for each lake surface elevation profile, we used robust MAD statistical method to exclude the spurious elevation measurements, possibly induced by land contamination So, in our evaluation, the influence of surrounding topography has been minimized. Considering the current length of this manuscript, in this revision we referred the reader to the following two papers for a more detailed discussion of the influence of surrounding topography:

Jiang, L., Nielsen, K., Dinardo, S., Andersen, O.B., Bauer-Gottwein, P. (2020). Evaluation of Sentinel-3 SRAL SAR altimetry over Chinese rivers. Remote Sensing of Environment, 237, 111546 Biancamaria, S., Schaedele, T., Blumstein, D., Frappart, F., Boy, F., Desjonqueres, J.D., Pottier, C., Blarel, F., Nino, F. (2018). Validation of Jason-3 tracking modes over French rivers. Remote Sensing of Environment, 209, 77-89

- In the conclusions, the FF-SAR could be cited for future investigations (see https://doi.org/10.1016/j.rse.2019.111589 ) as services will be providing FF-SAR Sentinel-3 and Cryosat-2 data shortly (Scagliola et al. 2020 in OSTST2020, Moreau et al. in the 2020 Coastal Altimetry Workshop final report available at http://doi.org/10.5270/esa.caw12$_2020.final_report.).Sentinel-$ $6datacanalsobeprocessedinFullyFocusedmodewhenavailable.$

Response: As the reviewer suggested, we have cited the paper in the conclusion section and remarked that it would be a worthy direction for future investigation.

Specific comments Abstract section: For Sentinel-3, Tables 9 and 8 indicate that the mean results are equivalent for the OCOG and SAMOSA, this should be underlined in the discussion. The bias (Table 7) is way lower for Sentinel-3. Therefore, the statement in the abstract ("The results show that the model-free retrackers (e.g. OCOG/Ice- 1/Ice) outperform the model-based retrackers for all missions, particularly over small lakes.") shall be revised.

Response: Following the reviewer's suggestion, we have underlined the equivalent performances of these two retrackers in discussion. We have also revised the sentence in the abstract to "The results show that the model-free retrackers (e.g. OCOG/Ice-1/Ice) outperform the model-based retrackers for most of the missions, particularly over small lakes"

Introduction: - In citing each mission, a reference paper should be added.

Response: Following the reviewer's suggestion, we have added the reference for the general information of each satellite mission.

- Please indicate that Cryosat-2 is able operating the SARin mode in: "Most of the radar altimeters operate in a conventional low-resolution mode (LRM), whereas Sentinel-3 and Cryosat-2 operate in Synthetic Aperture Radar (SAR) mode."

Response: Revised as the reviewer suggested.

- Please support the following with a reference: "the River and Lake database (http://www.cse.dmu.ac.uk/EAPRS/products$_{r}iverlake.html)builtbytheESAandDeMontfortUniversity(ESA-DMU),"asmadefortheotherdatabases.$

Response: As the reviewer suggested, the most relevant paper has been cited in the revised manuscript for this database.

- The position of this reference: "Jarihani et al. (2013) compared five different satellite [. . .]" shall be revised in the references list as the name is reported before the surname: Asadzadeh Jarihani, A., Callow, J. N., Johansen, K., and Gouweleeuw, B.: Evaluation of multiple satellite altimetry data for studying inland water bodies and river floods, Journal of Hydrology, 505, 78-90, 10.1016/j.jhydrol.2013.09.010, 2013.

Response: Revised. Thanks for the careful reading.

- Please evaluate revising from "self-developed retrackers" to "non-official retrackers" when citing Villadsen et al. (2016). Please cite also this paper in the sentence: https://doi.org/10.1016/j.rse.2019.111546.

Response: The sentence has been changed following the reviewer's suggestion. And the paper has been cited in the revised manuscript.

- The following could be a bit more detailed: "HY-2A was excluded from this study because of the difficulty in obtaining its data product."

Response: As the reviewer suggested, we revised the sentence as " HY-2A was excluded from this study because of the difficulty in obtaining its data product (The data is not available for the public)"

Section 2.1 - The overall discussion on ice cover and presence of small islands is fine. Can something more be said about the complexity of the topography surrounding each of the investigated lake? This should be related to the tracking modes (open loop/closed loop) size of the receiving window of the specific altimetry system to enhance the discussion. This is a very important point which is not discussed in detail in the paper (e.g. ENVISAT operated with 3 possible bandwidths/receiving window sizes allowing the instrument to correctly operate on various surfaces). This could be related to the data loss rate discussed in Table 6.

Response: We appreciate the reviewer's insightful comments. In response to the reviewer's comments, we added that the surrounding topography could have nonnegligible influences on elevation measurements in Section 4.2. Since the smallest case study lake in our evaluation is Reservoir Lokka in Finland with a surface area of about 500 km2. For each mission, the ground track over the lake is at least 10 kilometers long. To eliminate the possible influence of surrounding topography and land contamination. We have removed the observations within 2 km buffer distance from the shoreline. In addition, for each lake surface elevation profile, we used robust MAD statistical method to exclude the spurious elevation measurements, possibly induced by land contamination. So, we are confident that our evaluation results are minimally influenced by the surrounding topography.

We also include some information about Jason-3 and Sentinel-3 that have open-loop mode. ENVISAT operated with 3 different receiving window size. Considering the length of this manuscript, we did not include an in-depth discussion on these issues.

In this study, the "data loss rate" refers to the data loss rate of lake level estimates, instead of data loss rate of original elevation measurements. We have added two sentences in the result section to clarify this confusion and modified the abstract and conclusion accordingly.

Section 3 - Table 2 is introduced with the following sentence: "We used the most upto-date version of data product of each mission for the evaluation. The geographical coverage, operational time period, repeat cycle, footprint size and retrackers of these radar altimetry missions are summarized in Table 2.'. Please correct from "footprint size" to posting rate:

Response: Thanks for the careful reading. we changed the "footprint size" to "sampling rate".

- Please improve including 'empirical' 'physical' in the following sentence: "These retrackers can be divided into two general categories: the empirical/model-free retrackers and the physical/model-based retrackers." - Typo, plural in "retrackers": "and the Sentinel-3 Ice-Sheet retrackers is based on a 5-part piecewise analytical function (MSSL/UCL/CLS, 2019)."

Response: Revised as the reviewer suggested.

- Regarding "Jason-3 now operates on the nominal orbit and will continue until the planned launch of Jason-CS/Sentinel-6 in 2020." Please replace "Jason-CS/Sentinel-6" with "Sentinel-6 Michael Freilich"

Response: Changed as the reviewer suggested.

Section 4 - Typo "e" in : "and the most recent release e of the altimetry"

Response: Corrected.

- Regarding "Third, the ice-cover condition is examined using the simultaneous TB measurements from the MWR instruments, and those lake water level estimates during the ice-covered period are excluded in the subsequent accuracy evaluations." Did author consider the possibility of comparing TB measurements results to ice charts?

Response: We thank the reviewer's comments and suggestion. It would be helpful if the reviewer can let us know where we can find the ice charts.

Section 4.1 - Please indicate "orthometric height" in "Geoid converts the reference

surface from 330 ellipsoid to geoid(orthometric height)".

Response: Added as the reviewer suggested.

Section 6 - Regarding "Our evaluation result is contrary to Sulistioadi (2015), who found comparable performances between Sea Ice and OCOG retrackers over a couple of small lakes using ENVISat data." Please do not be generic and clearly name the lakes studied in Sulistioadi et. al (2015). As previously indicated, one cannot exclude that other factors (e.g. topography) filtering criteria played a role in justifying the results obtained by Sulistioadi et. al (2015). To confirm that the OCOG is better, your analysis should be done over the same lakes and the methodologies adopted compared in discussing the results.

Response: Following the reviewer's suggestion, we added the names of the lakes (Lake Matano and Lake Towuti in Indonesia) in order to clarify the confusion.

- A possible strategy to create a multi mission time series is discussed. If implemented, it would have added a significant contribution to the paper. - Regarding: "When a lake was visited by more than one satellite missions on the same day, the best water level estimate among the overlapping missions should be selected to form a long-term series of records, in terms of the performance (r and RMSE) of the missions", which criteria would authors suggest to select the "the best water level estimate among the overlapping missions"?

Response: We added the criteria as the reviewer suggested. The water level estimates from the satellite mission with higher r value and lower RMSE will be used.

- Typo (double full stop)in: "[…] in terms of the performance (r and RMSE) of the missions.."

Response: Deleted

- Please modify, according to table 9, from "6.47" to "6.08" in "The mean RMSE decreases from 35.17 cm of the early ERS-1 mission to 6.47 cm of the current Sentinel-3

mission."

Response: We thank the reviewer for the careful reading. We have revised the number.

On Tables Figures Table 2 - Sentinel-3 is indicated with a single launch date. Please consider including 2 entries for both Sentinel-3A Sentinel-3B.

Response: As the reviewer suggested, the launch dates of these two satellites have been added in Table 2.

Table 4 - Please explain how (see Sentinel-3 for example) for 18 Cycles you have 272 ground tracks selected for the first lake in the table.

Response: Table 4 lists the index of the ground track (ground track number) selected for the evaluation, not the total number of ground tracks. We have revised the caption of Table 4 to clarify the confusion.

Please also note the supplement to this comment:
https://hess.copernicus.org/preprints/hess-2020-510/hess-2020-510-AC1-supplement.pdf

---

## Author Comment (AC2) · 19 Jan 2021

Our Response to Anonymous Referee #2

-The paper evaluates water levels based on almost all historic and current altimetry missions and their standard retrackers over 12 lakes of different sizes. Here, especially, the results of the older missions are interesting. The main issue with this paper is the small sample size. 12 lakes are too small to provide any solid recommendations. Having a larger and more representative sample size would make this paper much more valuable. The Paper is well written and organized. The paper can be accepted if the review comment is addressed. Here, especially a discussion of low sample size is

[Figure]

Creative Commons CC-BY license logo

needed and the conclusions should be modified accordingly.

RESPONSE: We thank the reviewer for the positive comments on the value, writing and organization of our manuscript. We will address the sample size of case study lakes and other issues in the following item-by-item responses.

-General comments: To make solid statements and recommendations about the rekrak-ing performance, 12 lakes are too small a sample size. This should at least be mentioned in the discussions section. However, the results in the paper support similar results in the literature.

RESPONSE: The selection of case study samples lake for our evaluation must meet the two requirements: 1) the sample lakes must be overpassed by all the satellite missions; and 2) Simultaneous in situ gauge data are available for the sample lakes. After our thorough search, we have identified 12 sample lakes that satisfied these two conditions. In most of the previous similar evaluations (in the introduction section), usually less than 5 sample lakes were used in their evaluations, and 12 sample lakes for our evaluation study still represents the largest sample size in the literature. More importantly, the twelve lakes in our study are located in different continents, latitudes and geographical environments. They include both natural lakes and reservoirs. They have different sizes, and winter ice conditions. We believe that this group of sample case study lakes are representative for the majority of inland lakes around the world and therefore we are confident that evaluation results for the historical and operational satellite altimetry missions through these sample lakes are valid. Nevertheless, we agree that it is even better if we have a much larger sample size that satisfy the above conditions. In response to the reviewer's comments, we have added a brief discussion on the lake sample size in Section 2.1 in the revised manuscript, and we hope that we can include more sample lakes in our future research when their in-situ gauge data become available.

-The method section is vague and must be extended so it at least summarizes the
methods from the mentioned reference studies. Hence, The MAD is estimated but what is the threshold to reject an observation.

RESPONSE: In response to the reviewer's comments, we have added more information about the robust MAD statistical method in the revised manuscript. We have also clarified the threshold value of the MAD statistic score used to exclude an observation from the subsequent calculation. A main point in the paper is to construct consistent long-term time series and one of the issues is the intermission/retracking bias. In section 5.2 the gauge is used to estimate the biases. However, as discussed in the Discussion Section a gauge is not always available and therefore the bias should be estimated relative to a reference(s) mission. Why did the authors not test this approach? RESPONSE: We appreciate the reviewer's comment and suggestion. This primary purpose of this study is to evaluate the historic and operational missions, to identify the reference mission, and then to develop a general strategy for estimating the biases. The estimation of the biases and the construction of long-term records need to carefully consider the temporal and spatial overlapping between these missions, particularly the overlapping with the consistent Topex/Poseidon-Jason series. This entails much more work on data processing, result analysis and discussion. Based on the current work, we plan to construct consistent long-term time series at regional or global scale in the future, relative to a reference mission as the reviewer suggested.

-Why do the authors select evaluation targets in ice-covered regions when measurements during ice-covered periods are removed anyway?

RESPONSE: Lakes located in high latitude are more frequently overpassed by satellite missions, but the ice cover in the winter season may introduce significant errors to the elevation measurements of satellite altimetry missions. Since the official retrackers of all the satellite altimetry missions are not designed to handle the ice-cover on lakes, we identified and excluded the measurements obtained in the ice-covered condition in order to have a fair comparison between different altimetry missions. To clarify this confusion, we have added a brief discussion in Section 4.3 in the revised manuscript.

We also noted that a non-official retracker developed by Shu et al. (2020) is able to accurately retrieve the water-equivalent lake level in the ice-covered condition for constructing a seasonally consistent lake water level time series from Sentinel-3 altimetry observations.

-Why only use 1 track from each mission in the time series if more are available, this would improve the temporal resolution and the statistical foundation. Anyway, some of the missions are in different orbits anyway. For this reason, C2 could also have been included. Several authors have successfully applied C2 for lake level estimation.

RESPONSE: We appreciate the reviewer's comment. Indeed, including more ground tracks will increase the temporal resolution of the time series. This is recommended when the analysis of the temporal variation of lake water levels is the primary goal in the practical applications. Nevertheless, the primary focus of this study is to compare and evaluate the performances of multiple satellite radar altimetry missions. Our strategy is to minimize the influences of distant ground tracks from the gauge stations and ensure the objective comparison and evaluation of these missions. For a large lake (e.g. the Great Lakes), strong wind, big wave, diurnal tide, geoid undulation, and other factors may significantly influence lake water level at different locations in the lake. The in-situ water level measurements from a gauge station may not reflect the actual water level of those ground tracks far away from the gauge station. Thus, the overall RMSE of the altimetry-derived estimates will increase when altimetry observations from distant ground tracks are included for evaluation (Birkett, 1995). To minimize the possible influence of wind, waves, tide and other environmental factors for an objective comparison between different satellite missions, we thus select the ground track nearest to the gauge station and exclude distant ground tracks in the performance evaluation. As sufficient footprints along the nearest ground track are available, we are able to derive statistically reliable RMSE and r values for objective comparisons between different missions. Currently, CryoSat-2 uses a geodetic orbit (long-term repeat orbit). It is difficult to form a time series of co-located water level estimates for the evaluation. Although a time series of water level estimates from CryoSat-2 observations can be derived for a large lake by including many different ground tracks separated by a large distance, this will inevitably introduce the evaluation uncertainties due to the wind, wave, tide, and other environmental factors as explained above. This is why we did not include the Cryosat-2 data in the evaluation. For the same reason, we did not include the data collected by other satellite missions during their geodetic phases or drifting phases. In this revision, we have added a brief explanation why we only include the nearest ground track in our evaluation, and included the following reference: Birkett, C.M. (1995). The contribution of TOPEX/POSEIDON to the global monitoring of climatically sensitive lakes. Journal of Geophysical Research, 100, 25,179-"125,204"

Specific Comments L296: Shu et al, 2020 is not the reference of the standard S3A retrackers.

RESPONSE: In response to the reviewer's suggestion, the citation of (Shu et al, 2020) has been removed and the correct citation has been added.

L306: why only use such a small time period of S3 and Jason-3 in the evaluation?

RESPONSE: This research involves a large volume of data processing for all eleven missions over the twelve lakes. We started the data processing with Sentinel-3, then Jason-3. At the time we wrote this manuscript, the available data for Sentinel-3 and Jason-3 was from 2016 to 2018. Since the time period is longer than a full year (winter and summer), which cover a full hydrologic cycle of lakes. So, we believe that it is sufficient to support the performance evaluation of these two missions, although more data become available now.

L331: add a reference to EGM2008

RESPONSE: Added as the reviewer suggested.

L361: which criterion is used to remove outliers

RESPONSE: In the revised manuscript, we have added that the measurements with

MAD statistic score larger than and equal to 3 are excluded.

L364: "through" -> over

RESPONSE: Corrected.

L393: The r indicates -> the Pearson correlation r ...

RESPONSE: Changed according to the reviewer's suggestion.

L420: When you calculate the data loss rate is that based on the "valid" measurements or all measurements

RESPONSE: It is based on the valid measurements after filtering the spurious measurements. In this study, the "data loss rate" refers to the data loss rate of lake level estimates, instead of data loss rate of elevation measurements. We have added two sentences in the result section to clarify this confusion and modified the abstract and conclusion accordingly.

L440: This only makes sense to state if the gauge and altimetry has the same vertical reference

RESPONSE: We agree. We modified the sentence to clarify this issue.

L448: is the bias calculated w.r.t the gauge? then add this

RESPONSE: Yes. We added this information in the revised manuscript.

L495-503: Put all the numbers in a table

RESPONSE: These numbers are summarized in Table 9. The numbers are cited here for the description purpose.

L510: Such conclusions are difficult to state based on just a few lakes

RESPONSE: As responded above, the selection of case study samples lake is limited by two requirements: 1) the sample lakes must be overpassed by all the satellite missions; and 2) Simultaneous in situ gauge data are available for the sample lakes. In most of the previous similar evaluations, usually less than 5 sample lakes were used in their evaluations. The 12 sample lakes in this study still represents the largest sample size in the literature. More importantly, the twelve lakes in our study are located in different continents, latitudes and geographical environments. They include both natural lakes and reservoirs. They have different sizes, and winter ice conditions. We believe that this group of sample case study lakes can represent the majority of inland lakes around the world and therefore we are confident that evaluation results for the historical and operational satellite altimetry missions through these sample lakes are valid. Nevertheless, we agree that it is even better if we have a much larger sample size that satisfy the above conditions. In response to the reviewer's comments, we have added a brief discussion on the lake sample size in Section 2.1 in the revised manuscript, and we hope that we can include more sample lakes in our future research when their in-situ gauge data become available.

L582: How would you determine which mission provides the best measurement?

RESPONSE: In the revised manuscript, we added that the water level estimates from the satellite mission with higher r and lower RMSE should be used.

Please also note the supplement to this comment:
https://hess.copernicus.org/preprints/hess-2020-510/hess-2020-510-AC2-supplement.pdf

---

## Author Response (AR2)

**Our Response to the Editor's Comments**

Comments to the Author:

Dear authors,

Thank you for providing your revised manuscript and your thorough response to reviews.

I have two minor revisions I would like to suggest:

-I'd like to encourage you to broaden your last paragraph, either by revising it or by adding a few more lines. It seems like the issue of small sample size is a general problem faced in all of this type of work. Beyond just pointing to this as a limitation of this study, I'd encourage you to add a few words to describe the greater problem faced by studies such as yours in terms of sample size. This is somewhat alluded to in the first few lines of that paragraph, but I'd encourage you to really make this point more generally in this paragraph.

**RESPONSE:** We appreciate this comment. We have revised the last paragraph and pointed out that the small lake sample size is a general issue for similar studies that evaluate the performances of multiple satellite missions.

-On line 652, you state that 12 lakes represents the "largest sample size", but did not contextualize what was meant by this. Could you include additional references here and give a little more context in terms of what you are comparing to?

**RESPONSE:** Following this suggestion, in the revised manuscript, we have added a description on the lake sample size in previous similar studies as a comparison, to clarify the confusion.

I'd also encourage one last thorough read for small grammatical errors.

**RESPONSE:** We thank the editor for the very careful reading. We have done another round of thorough read of this manuscript and corrected the grammatical errors that we can identify.

I look forward to reviewing your updated manuscript.

Christa